# Conditional protein tagging methods reveal highly specific subcellular distribution of ion channels in motion-sensing neurons

Sandra Fendl[1,2†*], Renee Marie Vieira[1†], Alexander Borst[1,2]

[1]Max Planck Institute of Neurobiology, Martinsried, Germany; [2]Graduate School of Systemic Neurosciences, LMU Munich, Martinsried, Germany

**Abstract** Neurotransmitter receptors and ion channels shape the biophysical properties of neurons, from the sign of the response mediated by neurotransmitter receptors to the dynamics shaped by voltage-gated ion channels. Therefore, knowing the localizations and types of receptors and channels present in neurons is fundamental to our understanding of neural computation. Here, we developed two approaches to visualize the subcellular localization of specific proteins in *Drosophila*: The flippase-dependent expression of GFP-tagged receptor subunits in single neurons and 'FlpTag', a versatile new tool for the conditional labelling of endogenous proteins. Using these methods, we investigated the subcellular distribution of the receptors GluClα, Rdl, and Dα7 and the ion channels para and Ih in motion-sensing T4/T5 neurons of the *Drosophila* visual system. We discovered a strictly segregated subcellular distribution of these proteins and a sequential spatial arrangement of glutamate, acetylcholine, and GABA receptors along the dendrite that matched the previously reported EM-reconstructed synapse distributions.

**\*For correspondence:**
sfendl@neuro.mpg.de

†These authors contributed equally to this work

## Introduction

How neural circuits implement certain computations in order to process sensory information is a central question in systems neuroscience. In the visual system of *Drosophila*, much progress has been made in this direction: numerous studies examined the response properties of different cell-types in the fly brain and electron microscopy studies revealed the neuronal wiring between them. However, one element crucial to our understanding is still missing; these are the neurotransmitter receptors used by cells at the postsynaptic site. This knowledge is essential since neurotransmitters and corresponding receptors define the sign and the time-course of a connection, that is whether a synapse is inhibitory or excitatory and whether the signal transduction is fast or slow. The same neurotransmitter can act on different receptors with widely differing effects for the postsynaptic neuron. Glutamate for instance is mainly excitatory, however, in invertebrates it can also have inhibitory effects when it acts on a glutamate-gated chloride channel, known as GluClα (*Cully et al., 1996*; *Liu and Wilson, 2013*; *Mauss et al., 2015*). Recently, it has also been shown that acetylcholine, usually excitatory, might also be inhibitory in *Drosophila*, if it binds to the muscarinic mAChR-A receptor (*Bielopolski et al., 2019*). Hence, knowledge inferring the type of transmitter receptor at a synapse is essential for our understanding of the way neural circuits process information.

Moreover, voltage-gated ion channels shape synaptic transmission and the integration of synaptic inputs by defining the membrane properties of every neural cell type. The voltage-gated calcium channel cacophony, for instance, mediates influx of calcium ions that drives synaptic vesicle fusion at presynaptic sites (*Kawasaki et al., 2004*; *Fisher et al., 2017*). Voltage-gated sodium channels like paralytic (para) are important for the cell's excitability and the generation of sodium-dependent

action potentials. The voltage-gated channel Ih influences the integration and kinetics of excitatory postsynaptic potentials (*Magee, 1999*; *Littleton and Ganetzky, 2000*; *George et al., 2009*). However, only little is known about how these channels are distributed in neurons and how this shapes the neural response properties.

One of the most extensively studied neural circuits in *Drosophila* is the motion vision pathway in the optic lobe and the underlying computation for direction-selectivity. The optic lobe comprises four neuropils: lamina, medulla, lobula, and lobula plate (*Figure 1A*). As in the vertebrate retina, the fly optic lobe processes information in parallel ON and OFF pathways (*Joesch et al., 2010*; *Borst and Helmstaedter, 2015*). Along the visual processing chain, T4/T5 neurons are the first neurons that respond to visual motion in a direction selective way (*Maisak et al., 2013*; *Behnia et al., 2014*; *Fisher et al., 2015a*; *Arenz et al., 2017*; *Strother et al., 2017*). T4 dendrites reside in layer 10 of the medulla and compute the direction of moving bright edges (ON-pathway). T5 dendrites arborize in layer 1 of the lobula and compute the direction of moving dark edges (OFF-pathway) (*Maisak et al., 2013*). The four subtypes of T4/T5 neurons (a, b, c, d), project axon terminals to one of the four layers in the lobula plate, each responding only to movement in one of the four cardinal directions, their preferred direction (*Maisak et al., 2013*).

How do T4/T5 neurons become direction-selective? Both T4 and T5 dendrites span around eight columns collecting signals from several presynaptic input neurons, each of which samples information from visual space in a retinotopic manner (*Haag et al., 2016*; *Shinomiya et al., 2019*). The functional response properties of the presynaptic partners of T4/T5 have been described in great detail (*Behnia et al., 2014*; *Ammer et al., 2015*; *Fisher et al., 2015a*; *Fisher et al., 2015b*; *Serbe et al., 2016*; *Arenz et al., 2017*; *Strother et al., 2017*; *Strother et al., 2018*; *Drews et al., 2020*) along with their neurotransmitter phenotypes (*Takemura et al., 2017*; *Richter et al., 2018*; *Shinomiya et al., 2019*; *Davis et al., 2020*). T4 dendrites receive glutamatergic, GABAergic and cholinergic input, whereas T5 dendrites receive GABAergic and cholinergic input only. These input synapses are arranged in a specific spatial order along T4/T5 dendrites (s. *Figure 1C and D*; for overview *Takemura et al., 2017*; *Shinomiya et al., 2019*).

Which receptors receive this repertoire of different neurotransmitters at the level of T4/T5 dendrites? Recently, several RNA-sequencing studies described the gene expression pattern of nearly all cell-types in the optic lobe of the fruit fly including T4/T5 neurons (*Pankova and Borst, 2016*; *Konstantinides et al., 2018*; *Davis et al., 2020*; *Hörmann et al., 2020*). T4/T5 neurons were found to express numerous receptor subunits of different transmitter classes and voltage-gated ion channels at various expression strengths. However, RNA-sequencing studies do not unambiguously answer the above question for two reasons: mRNA and protein levels are regulated in complex ways via post-transcriptional, translational, and protein degradation mechanisms making it difficult to assign protein levels to RNA levels (*Vogel and Marcotte, 2012*). Secondly, standard RNA-sequencing techniques cannot provide spatial information about receptor localizations, hence, they are not sufficient to conclude which transmitter receptors receive which input signal. Both shortcomings could in principle be overcome by antibody staining since immunohistochemical techniques detect neurotransmitter receptors at the protein level and preserve spatial information. However, high-quality antibodies are not available for every protein of interest and may have variable affinity due to epitope recognition (*Fritschy, 2008*). Furthermore, labeling ion channels via antibodies and ascribing expression of a given channel to a cell-type in dense neuronal tissue remains challenging. The disadvantages of the above techniques highlight the need for new strategies for labeling neurotransmitter receptors in cell types of interest.

In this study, we employed existing and generated new genetic methods to label and visualize ion channels in *Drosophila*. For endogenous, cell-type-specific labeling of proteins, we developed a generalizable method called FlpTag which expresses a GFP-tag conditionally. Using these tools, we explored the subcellular distribution of the glutamate receptor subunit GluClα, the acetylcholine receptor subunit Dα7, and the GABA receptor subunit Rdl in motion-sensing T4/T5 neurons. We found these receptor subunits to be differentially localized between dendrites and axon terminals. Along the dendrites of individual T4/T5 cells, the receptor subunits GluClα, Rdl, and Dα7 reveal a distinct distribution profile that can be assigned to specific input neurons forming synapses in this area. Furthermore, we demonstrated the generalizability of the FlpTag approach by generating lines for the metabotropic GABA receptor subunit Gaba-b-r1 and the voltage-gated ion channels para

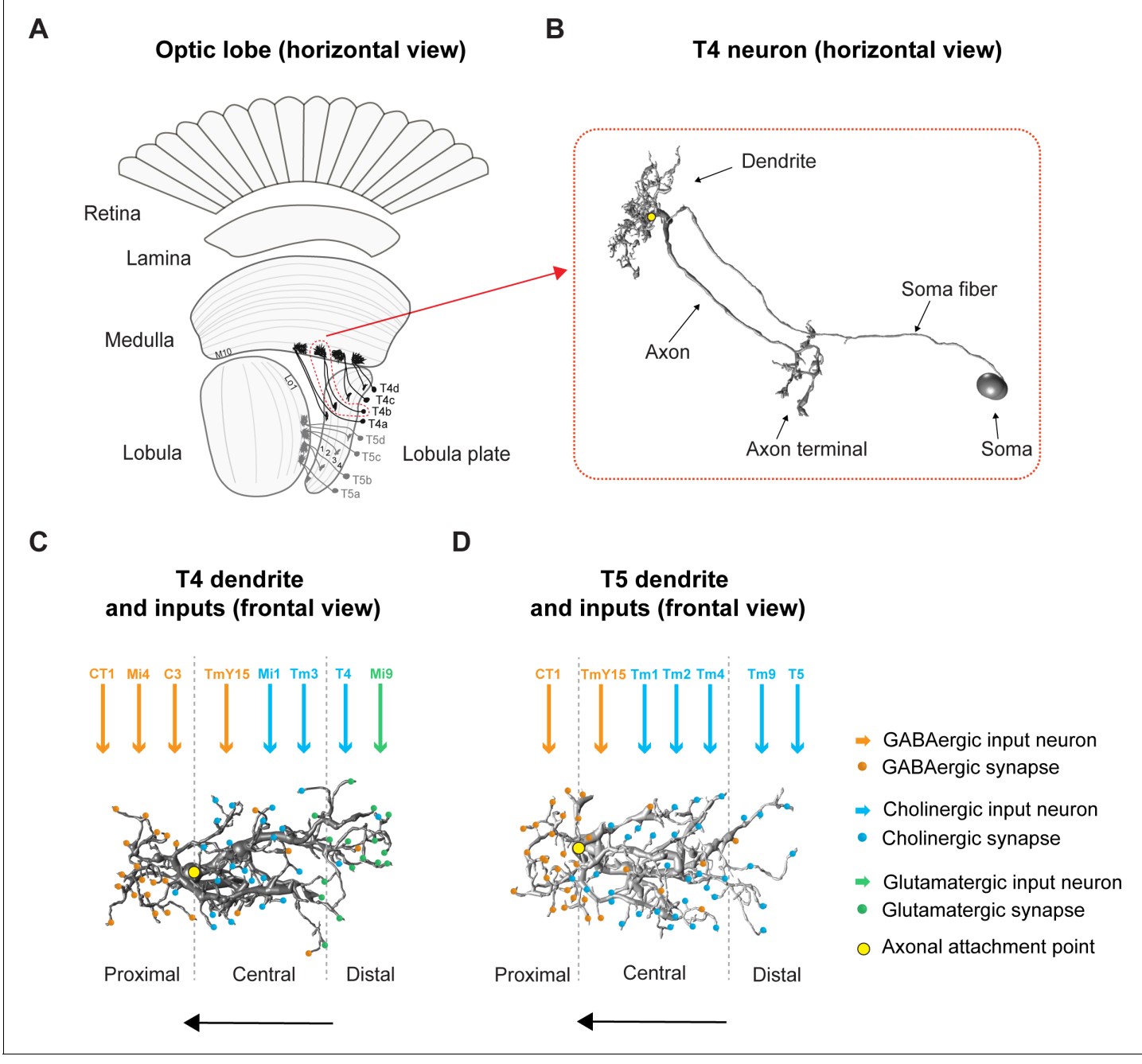

**Figure 1.** Overview of the fly optic lobe and anatomy of T4/T5 neurons with their presynaptic partners and distribution of input synapses. (**A**) Horizontal view of optic lobe with retina, lamina, medulla, lobula, and lobula plate. T4 dendrites (darker gray) reside in layer 10 of the medulla, T5 dendrites (lighter gray) in layer 1 of the lobula. T4/T5 axon terminals of all subtypes (a, b, c, d) project to the lobula plate in four layers. (**B**) Close-up, horizontal view of EM-reconstructed single T4 neuron with dendrite, axon, axon terminal, soma fiber and soma (image extracted from Seven medulla column connectome dataset, https://emdata.janelia.org/#/repo/medulla7column, #3b548, Janelia Research Campus). (**C**) Scheme of individual T4 dendrite and distribution of input synapses (frontal view). The dendrite depicted here is oriented pointing to the right side against its preferred direction from right to left (indicated by arrow). Input on proximal base of T4 dendrite: GABAergic CT1, Mi4 and C3. In the central area: GABAergic TmY15 and cholinergic Mi1 and Tm3. On the distal tips T4 receive input from cholinergic T4 from the same subtype and glutamatergic Mi9. Yellow circle labels first branching point of the dendritic arbor. Reproduced from *Figure 4*, *Shinomiya et al., 2019*, eLife, published under the Creative Commons Attribution 4.0 International Public License (CC BY 4.0; https://creativecommons.org/licenses/by/4.0/). (**D**) Scheme of individual T5 dendrite and distribution of input synapses (frontal view). The dendrite depicted here is oriented pointing to the right side against its preferred direction from right to left (indicated by arrow). The T5 dendrite receives GABAergic input from CT1 on the proximal base and from TmY15 in the central area. Cholinergic synapses are formed with Tm1, Tm2, and Tm4 in the central area and with Tm9 and T5 from the same subtype on the distal dendritic tips. Yellow circle labels first branching

*Figure 1 continued on next page*

and Ih. The strategies described here can be applied to other cells as well as other proteins to reveal the full inventory and spatial distribution of the various ion channels within individual neurons.

## Results

### Subcellular localization of the inhibitory glutamate receptor GluClα in T4/T5 neurons

As suggested by the connectome (*Takemura et al., 2017*; *Shinomiya et al., 2019*) and antibody staining against the vesicular glutamate transporter VGluT (*Richter et al., 2018*), T4 cells receive input on their dendrites from the glutamatergic medulla neuron Mi9. Since a multitude of glutamate receptors exist, both excitatory and inhibitory, we explored which glutamate receptor forms the synapse between the glutamatergic Mi9 input and T4 dendrites.

According to a RNA-sequencing study, GluClα is the most highly expressed glutamate receptor in T4 neurons (*Davis et al., 2020*). To investigate the distribution of this glutamate receptor in T4 and T5 neurons, we developed a transgenic fly line that allowed us to express a GFP-tagged GluClα in a cell-type specific way. We created a *UAS-GluClα::GFP* line bearing the cDNA of GluClα with a GFP-insertion (*Supplementary file 1*). This construct can be combined with any *Gal4*-line to study the receptor's expression and its subcellular localization. We combined the *UAS-GluClα::GFP* line with a membrane-bound *UAS-myr::tdTomato* and expressed both constructs under the control of a T4/T5-specific *Gal4*-driver line. We found GluClα in T4 dendrites of the medulla, where it is distributed in discrete puncta (*Figure 2A*; horizontal section, first two panels). A top view of the medulla of these flies reveals that these puncta are arranged in circular clusters, each corresponding to one column (*Figure 2A*, right panel). Since Mi9 is the only glutamatergic presynaptic partner of T4 cells in the medulla (*Takemura et al., 2017*; *Richter et al., 2018*; *Shinomiya et al., 2019*), this columnar arrangement likely reflects the columnar array of Mi9 cell inputs. Conversely, T5 dendrites are completely devoid of GluClα signal (*Figure 2A*, first two panels). This result is in agreement with T5 dendrites not receiving glutamatergic input (*Richter et al., 2018*). In addition to the medulla layer 10, GFP signal of GluClα::GFP is also visible in the axon terminals of T4/T5 in the lobula plate (*Figure 2A*, first two panels). However, both T4 and T5 cells send their axons into the lobula plate, therefore, this staining cannot be assigned to one of the cell types specifically. To differentiate between the two cell types, we used two different driver lines, one specific for either T4 or T5 cells. We confirmed the presence of GluClα in the dendritic layer of T4 cells (*Figure 2B*) and the lack thereof in the dendritic layer of T5 cells (*Figure 2C*). Interestingly, with these specific driver lines, both T4 and T5 neurons express the glutamate receptor in their axon terminals in the lobula plate (*Figure 2B* and *Figure 2C*). The presence of GluClα in the axon terminals of T5 neurons explains the high GluClα-mRNA levels in T5 (*Davis et al., 2020*) even though T5 dendrites are missing a glutamatergic presynaptic partner (*Takemura et al., 2017*; *Richter et al., 2018*; *Shinomiya et al., 2019*).

One caveat associated with overexpression-lines is a potential mis-localization of proteins. To control for this effect, we used a pan-neuronal *Gal4-line* to express the *UAS-GluClα::GFP* construct and compared this expression pattern to an existing MiMIC protein trap line with GFP insertion (MiMIC GFSTF) in the endogenous locus of GluClα (Mi02890) (*Nagarkar-Jaiswal et al., 2015a*). We observed broad expression of GluClα throughout all neuropils of the optic lobe in both genotypes (*Figure 2—figure supplement 1A and B*). We quantified the mean fluorescence intensity of manually drawn ROIs around the medulla and found both values to be similar for the pan-neuronal *UAS-GluClα::GFP* and the MiMIC line (*Figure 2—figure supplement 1D*). Furthermore, we expressed the *UAS-GluClα::GFP* line with a driver line for T1, a cell-type which lacks GluClα mRNA (*Davis et al., 2020*). Our *UAS*-line confirmed this result as we could not detect significant levels of GluClα::GFP protein in T1 (*Figure 2—figure supplement 1E*). Hence, overexpression of GFP-tagged GluClα, introduced as a transgene, leads to a subcellular localization pattern that seems to be identical to the endogenous GluClα protein.

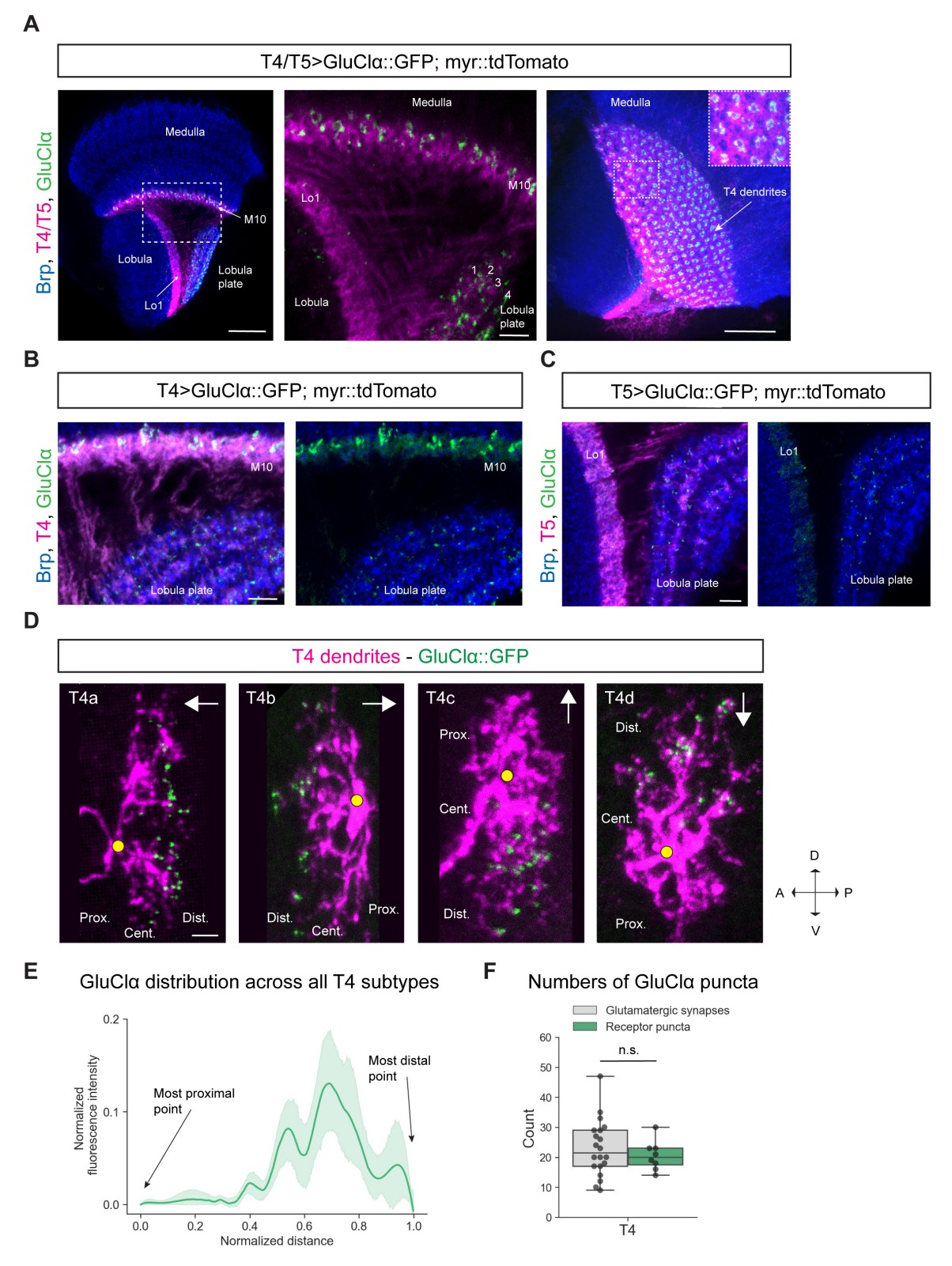

**Figure 2.** Subcellular localization of the inhibitory glutamate receptor GluClα in T4/T5 neurons. (**A**) Optic lobe with T4/T5 neurons labeled with myr:: tdTomato and GluClα::GFP. Left panel: horizontal view on the optic lobe overview (scale bar: 20 µm). Central panel: close-up of medulla layer M10, lobula layer Lo1 and lobula plate layers 1–4 (scale bar: 5 µm). Right panel: Frontal view on medulla layer M10 with T4 dendrites (scale bar: 20 µm); inset: close-up of columnar GluClα::GFP structure in layer 10 of the medulla. (**B**) Close-up of T4 dendrites in layer 10 of the medulla and axon terminals in

*Figure 2 continued on next page*

Figure 2 continued

lobula plate labeled with myr::tdTomato and GluClα::GFP (scale bar: 5 µm). (**C**) Close-up of T5 dendrites in layer 1 of the lobula and axon terminals in lobula plate labeled with myr::tdTomato and GluClα::GFP (scale bar: 5 µm). (**D**) Individual T4 dendrites labeled with tdTomtato and GluClα::GFP; subtypes a-d pointing in their natural orientation in visual space coordinates (A = anterior, p=posterior, D = dorsal, V = ventral). White arrows indicate preferred directions for every subtype and the dendrites' proximal (Prox.), central (Cent.) and distal (Dist.) areas are labeled (scale bar: 2 µm). Yellow circle labels first branching point of the dendrite. (**E**) Quantification of GluClα distribution over the whole dendritic length (normalized distance) averaged across several T4 dendrites from all subtypes (n = 8). All dendrites were aligned pointing to the right with the most proximal point at 0.0 and the most distal point at 1.0. (**F**) Quantification of GluClα puncta averaged across several T4 dendrites from all subtypes (mean ± SD = 20.5, 4.98 [n = 8]) (same cells used in E) compared to number of glutamatergic input synapses from Mi9 (mean ± SD = 23.0, 9.34 [n = 20]) (EM numbers: personal communication, K. Shinomiya, May 2020). n.s., not significant p>0.05 (p=0.37, t-test).

The online version of this article includes the following source data and figure supplement(s) for figure 2:

**Source data 1.** Table with numbers of GluClα puncta quantified for T4 dendrites.
**Figure supplement 1.** Pan-neuronal GluClα levels and distribution in the optic lobe are comparable for MiMIC GFSTF, FlpTag and *UAS*-line.

Given that Mi9 is the only glutamatergic input neuron to T4 dendrites and GluClα is the corresponding glutamate receptor, we hypothesized that GluClα should localize on the individual T4 dendrite exclusively where Mi9 makes glutamatergic synapses with the latter. Therefore, we wanted to visualize the distribution of GluClα at the single-cell level along individual T4 dendrites. The dendrites of each T4/T5 subtype are oriented pointing against their preferred direction (*Takemura et al., 2017*; *Shinomiya et al., 2019*). With respect to the point of axonal attachment to the dendrite, T4/T5 dendrites can be divided into a proximal, central and distal region (summarized in *Figure 1B–D*). Electron microscopy studies have shown that Mi9 forms synaptic contacts with T4 on the distal tips of its dendrite (*Figure 1C*; *Takemura et al., 2017*; *Shinomiya et al., 2019*). Since T4/T5 dendrites are strongly intermingled in their respective layers, it is not possible to resolve receptor localizations at the single-cell level by labeling the whole population. We used a flippase-based mosaic approach (*Gordon and Scott, 2009*) to sparsely label single T4/T5 neurons with tdTomato together with the *UAS-GluClα::GFP* construct. By using a *FRT-Gal80-FRT* with an hs-FLP, both *UAS-myr::tdTomato* and *UAS-GluClα::GFP* expression are dependent on the same stochastic FLP-event. A heat-shock-activated flippase removes the FRT-flanked *Gal4-repressor Gal80*, which disinhibits Gal4, promoting transcription of both *UAS*-reporters simultaneously resulting in expression of membrane-bound tdTomato and GFP-tagged GluClα in only a few cells of interest. In individual T4 dendrites, we observed that GluClα was predominantly localized to the distal tips, which holds true for all four T4 subtypes (*Figure 2D*). We quantified the intensity distribution of the GluClα::GFP-signal over dendritic distance in individual T4 dendrites. To combine and average this distribution for all four subtypes, we rotated dendrites from each subtype such that the proximal side was on the left side of the image and the distal tips were pointing to the right. Averaged intensities across all subtypes confirmed our observations on individual cells, showing that GluClα is indeed localized toward the distal dendritic tips of T4 dendrites (*Figure 2E*). In addition, we quantified the numbers of GluClα puncta for all subtypes and compared them to the synapse numbers of glutamatergic Mi9 inputs onto T4 determined by the previous EM study (*Shinomiya et al., 2019*). The number of GluClα-puncta per T4 cell dendrite (mean: 20.5 puncta) matches closely the number of glutamatergic input synapses made by Mi9 onto one T4 cell (mean: 23 synaptic contacts; personal communication, K. Shinomiya, May 2020) (*Figure 2F*). This suggests that every GluClα-punctum resolved by confocal microscopy in individually labeled T4 dendrites represents one postsynaptic GluClα receptor cluster corresponding to one Mi9-T4 synapse.

In summary, GluClα localizes to the dendrites of T4 cells and to the axon terminals of both T4 and T5 cells. At the single-cell level, GluClα is distributed toward the distal tips of the dendrites in all T4 subtypes. Strikingly, the number of GluClα puncta closely matches the number of input synapses provided by Mi9, the only glutamatergic input neuron to T4 dendrites.

## Rdl localizes to T4/T5 dendritic compartments receiving GABAergic input

Having identified glutamatergic synapses, we employed similar methods to visualize GABAergic synapses in T4/T5 neurons. T4 dendrites receive input from several GABAergic cell-types in the medulla: on the proximal base of the dendrite, these are the columnar cells Mi4, C3; the multicolumnar

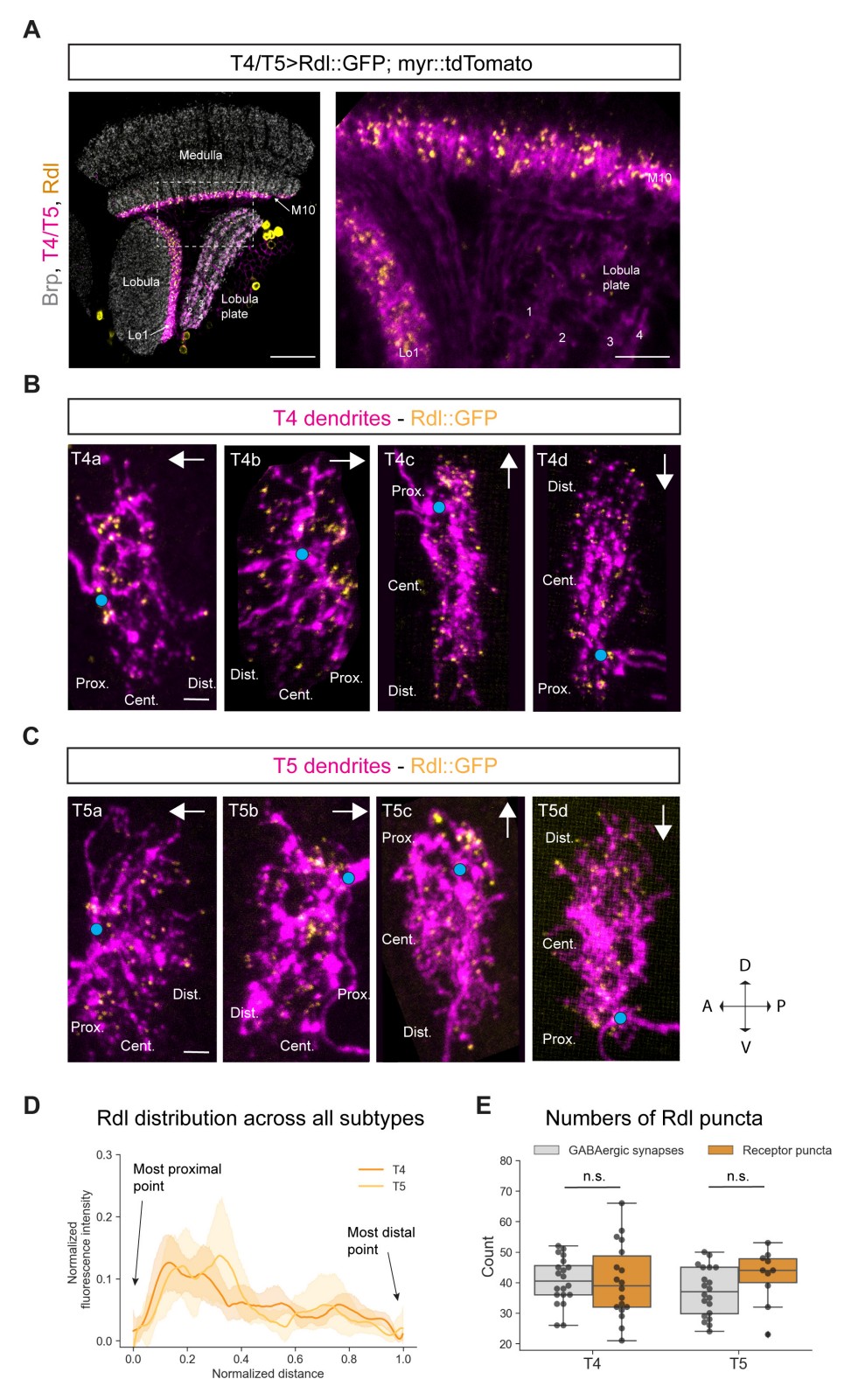

**Figure 3.** Subcellular localization of the GABA receptor Rdl in T4/T5 neurons. (**A**) Optic lobe with T4/T5 neurons labeled with myr::tdTomato and Rdl:: GFP. Left panel: horizontal view on the optic lobe overview (scale bar: 20 μm). Right panel: close-up of medulla layer M10, lobula layer Lo1 and lobula plate layers 1–4 (scale bar: 5 μm). (**B**) Individual T4 dendrites labeled with tdTomtato and Rdl::GFP; subtypes a-d pointing in their natural orientation in visual space coordinates (A = anterior, p=posterior, D = dorsal, V = ventral). White arrows indicate preferred directions for every subtype and the

*Figure 3 continued on next page*

Figure 3 continued

dendrites' proximal (Prox.), central (Cent.) and distal (Dist.) areas are labeled (scale bar: 2 µm). Blue circle labels first branching point of the dendrite. (**C**) Individual T5 dendrites labeled with tdTomtato and Rdl::GFP; subtypes a-d pointing in their natural orientation in visual space coordinates (A = anterior, p=posterior, D = dorsal, V = ventral). White arrows indicate preferred directions for every subtype and the dendrites' proximal (Prox.), central (Cent.) and distal (Dist.) areas are labeled (scale bar: 2 µm). Blue circle labels first branching point of the dendrite. (**D**) Quantification of Rdl distribution over the whole dendritic length (normalized distance) averaged across several T4 (n = 18) and T5 dendrites (n = 10) from all subtypes. All dendrites were aligned pointing to the right with the most proximal point at 0.0 and the most distal point at 1.0. (**E**) Quantification of Rdl puncta averaged across several T4 (mean ± SD = 40.4, 12.17 [n = 18]) and T5 dendrites (mean ± SD = 42.2, 8.88 [n = 10]) (same cells used in D) from all subtypes compared to number of GABAergic input synapses from T4 (mean ± SD = 40.5, 7.67 [n = 20]) and T5 (mean ± SD = 37.0, 8.05 [n = 20]) (EM numbers: personal communication, K. Shinomiya, May 2020). n.s., not significant p>0.05 (p=0.99 and p=0.13 respectively, t-test).

The online version of this article includes the following source data and figure supplement(s) for figure 3:

**Source data 1.** Table with numbers of Rdl puncta quantified for T4/T5 dendrites.

**Figure supplement 1.** Rdl is not detectable in the lamina neuron L1.

---

amacrine cell CT1 in the middle and distal part of the dendrite as well as TmY15 (*Figure 1C*). In contrast, T5 dendrites receive GABAergic input from only two cell-types: CT1 on the proximal base and TmY15 again throughout the central and distal area of the dendrite (*Figure 1D*). In total, T4 and T5 dendrites receive roughly the same number of GABAergic input synapses (*Takemura et al., 2017*; *Shinomiya et al., 2019*). Three ionotropic GABA receptor subunits are described in the *Drosophila* genome: Rdl, Lcch3, and Grd (*Liu et al., 2007*). We focused on the GABA receptor subunit Rdl, since RNA-sequencing studies had identified Rdl as the most highly expressed ionotropic GABA receptor subunit in T4 and T5 neurons (*Pankova and Borst, 2016*; *Davis et al., 2020*). Five Rdl subunits can form a homomeric chloride channel which leads to hyperpolarization upon GABA-binding, thus representing a receptor (*Ffrench-Constant et al., 1993*). Previous studies had created and used a UAS-Rdl::HA line to investigate the distribution of this GABA receptor subunit in *Drosophila* motoneurons and LPTCs (*Sánchez-Soriano et al., 2005*; *Raghu et al., 2007*; *Kuehn and Duch, 2013*). In our hands, the anti-HA staining of this line was too weak for conclusive results (data not shown), hence, we created a *UAS-Rdl::GFP* line, consisting of the coding sequence of Rdl and a GFP-tag (*Supplementary file 2*). Combining this line with a T4/T5 specific *Gal4*-line and a membrane-bound tdTomato revealed Rdl expression in both T4/T5 dendrites, but not in the axon terminals (*Figure 3A*). Taken together, both T4 and T5 neurons receive GABAergic inhibition via Rdl receptors on their dendrites.

In a control experiment, we tested for potential overexpression artifacts of the *UAS-Rdl::GFP* line. According to RNA-sequencing, Rdl is not expressed in the lamina monopolar neuron L1 (*Davis et al., 2020*). When we overexpressed *UAS-Rdl::GFP* by means of a *L1-Gal4* driver line, Rdl signal is not detectable in L1 dendrites (*Figure 3—figure supplement 1*). The Rdl::GFP protein was only visible in the cell bodies, presumably due to impaired protein translocation. This suggests that overexpressed Rdl only localizes to endogenous GABA synapses that are composed of the Rdl subunit. Hence, this line can be used to study the subcellular localization of Rdl in any given cell of interest.

Next, we looked at the distribution of the GABA receptor Rdl on individual T4 and T5 dendrites. Using the sparse labeling technique described above, we examined the Rdl::GFP distribution in individual T4/T5 dendrites. We found Rdl on the proximal base and in the central area of both T4 and T5 dendrites across all four subtypes (*Figure 3B* and *Figure 3C*). On the proximal base most of the Rdl-signal was arranged in strong discrete clusters, whereas sparse puncta localized to the central area and toward the distal tips. The strong Rdl-signal on the proximal base of the dendrite likely corresponds to the high number of GABAergic inputs provided by the following inputs: CT1, Mi4 and C3 for T4 (32.2 synapses) and CT1 for T5 (30.3 synapses) (personal communication, K. Shinomiya, May 2020). The sparsely distributed Rdl-puncta in the center and tips likely correspond to TmY15 inputs for both T4 and T5 dendrites. This distribution is recapitulated in the intensity quantification across all T4/T5 subtypes, with high Rdl intensity on the proximal side and lower signal in the central and distal area (*Figure 3D*). We quantified the numbers of Rdl receptor clusters in T4 and T5 dendrites and compared them to the sum of all GABAergic input synapses (Mi4, C3, CT1, TmY15 for T4 and CT1, TmY15 for T5) to T4/T5 mapped by EM studies. We found similar numbers of roughly 40 receptor clusters for both T4 and T5 which match the sum of all GABAergic input synapses to T4

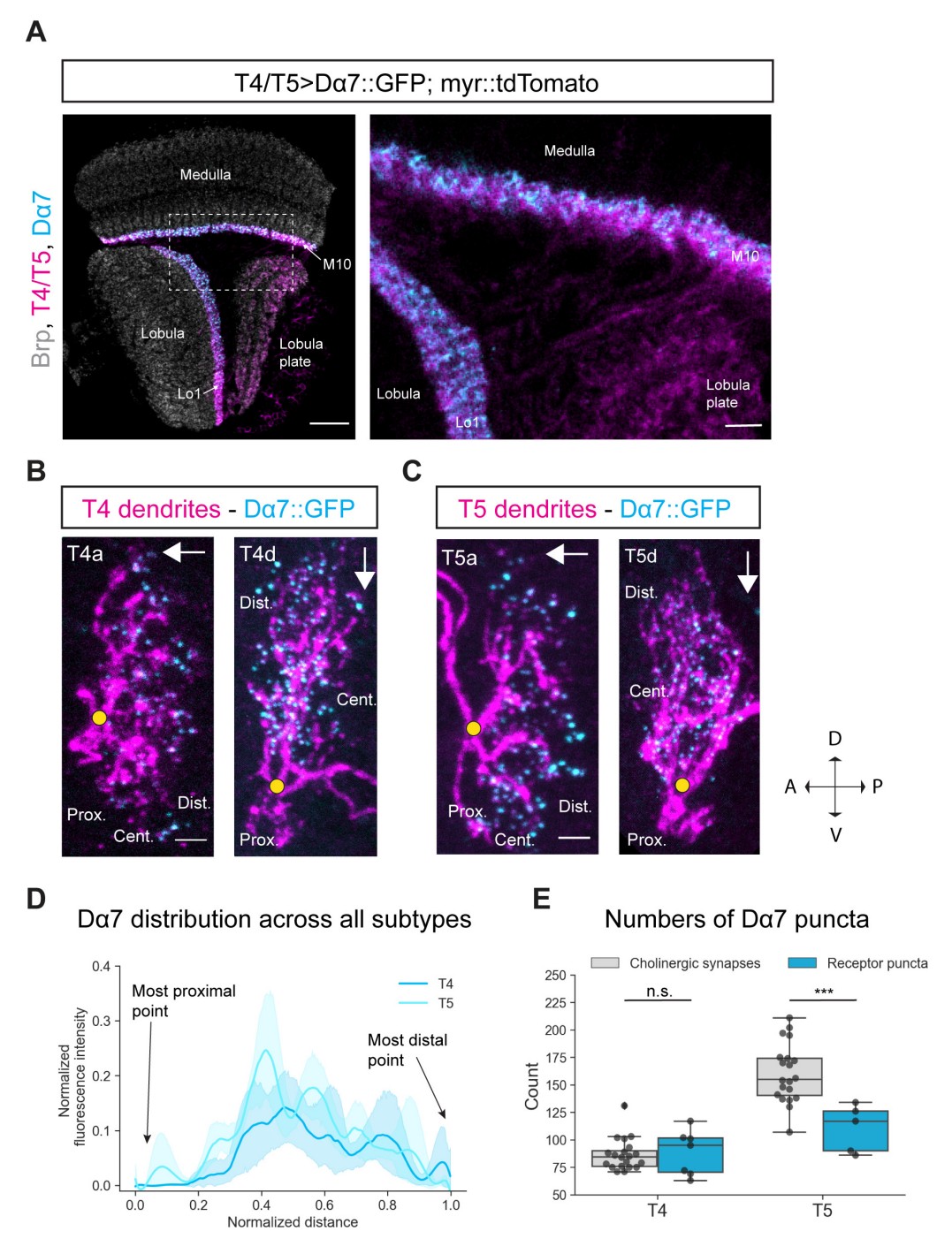

**Figure 4.** Subcellular localization of the ACh receptor subunit Dα7 in T4/T5 neurons. (**A**) Optic lobe with T4/T5 neurons labeled with myr::tdTomato and Dα7::GFP. Left panel: horizontal view on the optic lobe overview (scale bar: 20 μm). Right panel: close-up of medulla layer M10, lobula layer Lo1 and lobula plate layers 1–4 (scale bar: 5 μm). (**B**) Individual T4 dendrites labeled with tdTomtato and Dα7::GFP; subtypes a and d pointing in their natural orientation in visual space coordinates (A = anterior, p=posterior, D = dorsal, V = ventral). White arrows indicate preferred directions for every subtype and the dendrites' proximal (Prox.), central (Cent.) and distal (Dist.) areas are labeled (scale bar: 2 μm). Yellow circle labels first branching point of the dendrite. (**C**) Individual T5 dendrites labeled with tdTomtato and Dα7::GFP; subtypes a and d pointing in their natural orientation in visual space coordinates (A = anterior, p=posterior, D = dorsal, V = ventral). White arrows indicate preferred directions for every subtype and the dendrites' proximal (Prox.), central (Cent.) and distal (Dist.) areas are labeled (scale bar: 2 μm). Yellow circle labels first branching point of the dendrite. (**D**) Quantification of Dα7 distribution over the whole dendritic length (normalized distance) averaged across several T4 (n = 6) and T5 dendrites (n = 5) from all subtypes. All dendrites were aligned pointing to the right with the most proximal point at 0.0 and the most distal point at 1.0. (**E**) Quantification

*Figure 4 continued on next page*

*Figure 4 continued*

of Dα7 puncta averaged across several T4 (mean ± SD = 92.67, 18.67 [n = 6]) and T5 dendrites (mean ± SD = 110.6, 21.53 [n = 5]) (same cells like in D) from all subtypes compared to number of cholinergic input synapses for T4 (mean ± SD = 86.45, 14.37 [n = 20]) and T5 (mean ± SD = 160.50, 26.93 [n = 20]) (EM numbers: personal communication, K. Shinomiya, May 2020). n.s., not significant, p>0.05; ***p<0.001 (p=0.46 and p=2.1e-4 respectively, t-test).

The online version of this article includes the following source data and figure supplement(s) for figure 4:

**Source data 1.** Table with numbers of Dα7 puncta quantified for T4/T5 dendrites.
**Figure supplement 1.** Pan-neuronal Dα7 levels and distribution in the optic lobe as seen with *UAS-Dα7::GFP* line, Dα7 antibody staining and *Dα7-Trojan-Gal4* line.

---

(mean: 40.45) and T5 (mean: 37) (*Figure 3E*) (EM numbers: personal communication, K. Shinomiya, May 2020). Taken together, Rdl receptor subunits localize to the proximal base, and to a lesser extent, in the central area of the dendritic arbor of T4 and T5 neurons, reflecting their GABAergic inputs revealed by EM (*Shinomiya et al., 2019*).

## Dα7 localizes to T4/T5 dendritic compartments receiving cholinergic input

According to connectome data, T4 dendrites receive most of their input synapses from cholinergic Mi1 and Tm3 cells at the center of their dendrite (*Takemura et al., 2017*; *Shinomiya et al., 2019*). Furthermore, T4 neurons of the same subtype form synapses with each other at the distal tips of their dendrites (*Figure 1C*). As T4 neurons are cholinergic (*Mauss et al., 2014*; *Davis et al., 2020*), these T4-T4 synapses are thought to be cholinergic as well. With the exception of GABAergic CT1, T5 dendrites receive cholinergic input from Tm1, Tm2, and Tm4 in the central area of the dendrite. Tm9 and T5 provide cholinergic input mainly towards the distal tips of the dendrite (*Figure 1D*; *Takemura et al., 2017*; *Shinomiya et al., 2019*). T5 dendrites receive almost twice as many cholinergic inputs as T4; 160 and 87 synapses, respectively (*Shinomiya et al., 2019*). We used an existing GFP-tagged *UAS-Dα7::GFP* line to explore the subcellular distribution of these cholinergic synapses (*Raghu et al., 2009*). Dα7 is one of 10 different nicotinic ACh receptor subunits (Dα1-Dα7 and Dß1-Dß3) found in the *Drosophila* genome. All these subunits can form heteromeric receptors consisting of two or three subunits. In addition, Dα5, Dα6, and Dα7 can also form homomeric ACh receptors (*Lansdell and Millar, 2004*; *Lansdell et al., 2012*). According to RNA-sequencing data, both T4 and T5 neurons express almost every ACh receptor subunit, except for Dα6 and Dß3 (*Davis et al., 2020*). Expression of *UAS-Dα7::GFP* with a *T4/T5-Gal4* line, revealed the distribution of Dα7 to both T4 and T5 dendrites while their axon terminals remained devoid (*Figure 4A*).

As previously conducted, we tested for potential overexpression artifacts of the *UAS-Dα7::GFP* line. We expressed Dα7::GFP in all neurons and compared the expression pattern to two controls: first, an antibody staining against Dα7, and second, a MiMIC Trojan-*Gal4* (TG4) line for Dα7 combined with *UAS-Dα7::GFP* (*Figure 4—figure supplement 1A–C*; *Fayyazuddin et al., 2006*; *Diao et al., 2015*; *Lee et al., 2018*). The Trojan-*Gal4* (TG4) line has a *Gal4* insertion in the Dα7 gene, which drives expression of *Gal4* only under endogenous transcriptional control of Dα7. Combining this line with the reporter lines *UAS-myr::tdTomato* and *UAS-Dα7::GFP* should label all Dα7-expressing cells with tdTomato, and only within those cells, the Dα7 receptor subunits with GFP. In the pan-neuronal overexpression of *UAS-Dα7::GFP*, the ACh receptor subunit is broadly expressed throughout all neuropils with specific strong Dα7 signal in medulla layer 10 where T4 dendrites reside and lobula layer 1 where T5 dendrites reside (*Figure 4—figure supplement 1A*). However, in both the antibody- and the TG4-experiment, there is only weak Dα7 signal in M10 and Lo1 detectable (*Figure 4—figure supplement 1B and C*). Thus, under UAS-driven overexpression, the levels of Dα7 are increased compared to endogenous Dα7 levels in M10 and Lo1.

To assess whether the subcellular distribution of Dα7 is qualitatively altered by overexpression, we characterized the distribution of Dα7 in a cell type that does not express this receptor subunit endogenously. Transcriptomic data revealed that Dα7 is not expressed in Mi1 (*Davis et al., 2020*). However, Mi1 receives cholinergic input from L3 and L5 and expresses several different ACh receptor subunits (*Takemura et al., 2017*; *Shinomiya et al., 2019*; *Davis et al., 2020*). We tested the *UAS-Dα7::GFP* line in Mi1 to explore the qualitative overexpression-effects of this line. When *UAS-*

*Dα7::GFP* was overexpressed in Mi1, Dα7 localized to layers 1 and 5 of the medulla, where the dendrites of Mi1 neurons arborize and receive cholinergic input from L3 and L5 (*Takemura et al., 2017*; *Figure 4—figure supplement 1D*). This suggests that overexpressed Dα7::GFP localizes to cholinergic synapses and becomes part of an ACh-receptor, even if this subtype is not endogenously expressed in this neuron. If this scenario is true, the *UAS-Dα7::GFP* line does not report real endogenous subunit compositions with Dα7, but in general it can still be used as a marker for postsynaptic cholinergic sites.

To test this hypothesis, we performed sparse labeling of individual T4/T5 dendrites with the earlier described Gal80-hs-flippase method to explore the subcellular distribution of Dα7 along T4/T5 dendrites. Dα7 was distributed along the central area and distal tips of both T4 and T5 dendrites whereas the proximal base of the dendrite was completely devoid of Dα7 signal (*Figure 4B and C*). In the quantification, it becomes clear that for all subtypes the Dα7-intensity is strongest in the central area and slightly reduced toward the distal tips (*Figure 4D*). Taken together, these results demonstrate that with the *UAS-Dα7::GFP* line, Dα7 localizes to the areas where T4/T5 dendrites receive cholinergic input and not to the proximal base which receives only GABAergic synapses. We quantified the number of Dα7-puncta and compared it to the number of cholinergic synaptic contacts from T4/T5 inputs. For T4 dendrites the numbers of Dα7 puncta quantified (mean: 88.4) matched the numbers of cholinergic input synapses mapped by EM reconstruction (mean: 86.9; personal communication, K. Shinomiya, May 2020) (*Figure 4E*). This strongly suggests that Dα7 localizes only to cholinergic synapses. However, for T5 dendrites the Dα7 puncta exhibited 60 synapses less on average when compared to the mean of the summed cholinergic EM input synapse (*Figure 4E*). The levels of Dα7 along the dendrite are similar for T4 and T5 (*Figure 4D*), even though T5 receive more cholinergic inputs on their distal tips than T4 (*Shinomiya et al., 2019*). The main cholinergic input to T5 in the distal area is Tm9, which makes approximately 60 synapses with T5 dendrites. These 60 synapses could potentially be formed via different cholinergic receptors other than Dα7, for instance muscarinic ACh receptors (*Davis et al., 2020*).

In summary, the *UAS-Dα7::GFP* line cannot be used to define the exact composition of ACh receptor subunits of cholinergic synapses, but labels (nicotinic) ACh receptors in general. It, nevertheless, can be used as a marker for postsynaptic ACh receptors. Using this approach, we found that the central and distal areas of both T4 and T5 dendrites possess cholinergic receptors. The proximal base of the dendrites, as well as axon terminals are devoid of cholinergic input.

## FlpTag - a new tool for cell-type-specific, endogenous protein labeling

Additionally, we sought to observe the spatial distribution of endogenous receptors using a cell-type specific approach. We designed FlpTag, a new conditional, endogenous protein labeling strategy inspired by recently published flippase-dependent methods (*Fisher et al., 2017*; *Nagarkar-Jaiswal et al., 2017*; *Williams et al., 2019*).

The FlpTag cassette is a protein trap cassette consisting of a central GFP tag placed between a splice acceptor (SA) and splice donor (SD), flanked by specific Frt sites forming a FLEX-switch for stable inversion (*Figure 5A*, upper panel) (*Schnütgen et al., 2003*; *Xue et al., 2014*). The FlpTag cassette is integrated into an intronic coding region of interest by recombinase mediate cassette exchange (RMCE) in vivo. We used the existing intronic MiMIC gene trap with attP landing sites to facilitate ΦC31-dependent exchange of the MiMIC insertion with our FlpTag cassette, consisting of ΦC31 integrase attB sites on either end (*Venken et al., 2011*; *Nagarkar-Jaiswal et al., 2015b*). After ΦC31-dependent knock-in, two independent lines can be isolated. One in which the GFP is in the 5′ to 3′ direction; the same orientation as the gene. In this configuration FlpTag acts as a protein trap, revealing the protein's expression pattern. In the alternate orientation the FlpTag cassette is in the 3′ to 5′ direction; oppositely oriented to the gene. For the FlpTag approach, we used the oppositely oriented line in which the coding intron with the FlpTag cassette is naturally cut out during mRNA splicing and no labeling takes place. Only upon UAS-Gal4 driven, cell-type-specific expression of the Flp recombinase, the cassette is flipped in the same orientation as the gene. Due to the presence of flanking SA and SD, the GFP cassette is then spliced into the mature mRNA which is translated, labeling the protein with GFP (*Figure 5A*, lower panel).

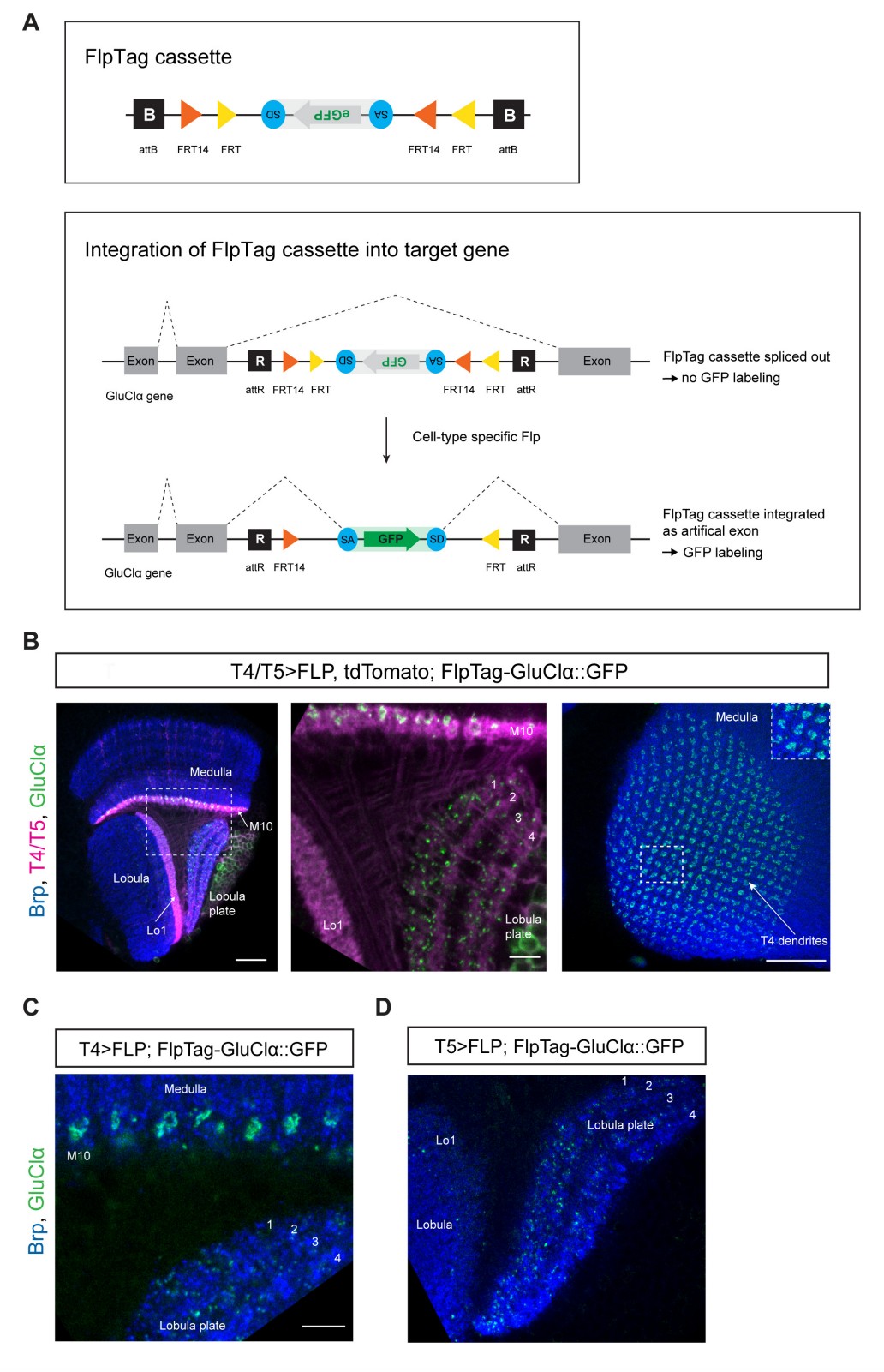

**Figure 5.** FlpTag, a new tool for cell-type-specific, endogenous labeling as shown with GluClα. (**A**) Scheme of FlpTag cassette (first panel) and integration of FlpTag cassette into target gene (second panel). The FlpTag cassette consists of attB-sites, specific FRT sites which form a FLEx-switch, a splice acceptor, GFP and a splice donor. After ΦC31-dependent integration of the FlpTag cassette into a coding intron of the GluClα target gene, two lines with opposite orientations of the cassette can be obtained. In the initial line with the cassette and GFP in opposite orientation with respect to the

*Figure 5 continued*

gene (shown here), the cassette is spliced out together with the intron and no GFP-labeling occurs. After cell-type-specific Flp expression, the FlpTag cassette is flipped, stably integrated as an artificial exon and GluClα is labeled with GFP. (B) Optic lobe with T4/T5 neurons labeled with myr::tdTomato and FlpTag-GluClα::GFP. Left panel: horizontal view on the optic lobe overview (scale bar: 20 µm). Central panel: close-up of medulla layer M10, lobula layer Lo1 and Lobula plate layers 1–4 (scale bar: 5 µm). Right panel: Frontal view on medulla layer M10 with T4 dendrites (scale bar: 20 µm); inset: close-up of columnar GluClα::GFP structure in layer 10 of the medulla. (C) Close-up of FlpTag-GluClα::GFP driven with a *T4-Gal4*-line; shown are layer 10 of the medulla where T4 dendrites reside and lobula plate layers 1–4 where T4 project their axon terminals to (scale bar: 5 µm). (D) Close-up of FlpTag-GluClα::GFP driven with a *T5-Gal4*-line; shown are layer 10 of the medulla where T4 dendrites reside and lobula plate layers 1–4 where T4 project their axon terminals to (scale bar: 5 µm).

## FlpTag line for GluClα

In a first proof-of-principle experiment, we generated a FlpTag line for the glutamate receptor subunit GluClα. The FlpTag cassette was inserted in the MiMIC insertion site MI02890, in the coding intron between the last two exons of the GluClα gene. For comparison of the various GluClα-tagged lines, we examined the expression patterns generated by pan-neuronal FlpTag-GluClα::GFP, MiMIC GFSTS GluClα, and pan-neuronal *UAS-GluClα::GFP*. The expression patterns were similar for all three lines (*Figure 2—figure supplement 1*). We combined the GluClα-FlpTag line with *UAS-FLPD.1* and a T4/T5-specific driver-line. The distribution pattern of GluClα seen here is virtually identical to the *UAS-GluClα::GFP* genotype: GluClα is localized to T4 dendrites, the T5 dendrite area is devoid of GluClα signal, and T4/T5 axon terminals in the lobula plate co-localize with GluClα (*Figure 5B*, compare with *Figure 2A*). Expression of flippase and FlpTag-GluClα in T4 neurons only further demonstrates the localization of the glutamate receptor to T4 dendrites and axon terminals, as seen before with the *UAS-GluClα::GFP* line (*Figure 5C*, compare with *Figure 2B*). Specific expression of flippase and FlpTag-GluClα in T5 neurons revealed that the receptor localizes specifically to the axon terminals in all T5 subtypes, as visualized by the presence of GluClα puncta in all layers of the lobula plate (*Figure 5D*, compare with *Figure 2C*).

Taken together, we generated a new *UAS*-line and developed a new tool for studying the localization of GluClα in a cell-type-specific manner. Both the *UAS-GluClα::GFP* line and the FlpTag-line led to similar results when compared to the pan-neuronal and T4/T5-specific experiments. These tools can be used interchangeably to study the subcellular localization of GluClα in any given cell of interest.

## FlpTag lines for Gaba-b-r1, para and Ih

The FlpTag approach is generalizable and can be applied to any of the >2800 fly lines available with MiMIC attP insertions in coding introns (*Nagarkar-Jaiswal et al., 2015b*). To demonstrate the universal applicability of our FlpTag strategy, we set out to generate more FlpTag lines with the aforementioned approach of integrating the FlpTag cassette into existing MiMIC landing sites in coding introns. In keeping with our interest in neurotransmitter receptors we explored another GABA receptor subunit, the metabotropic channel Gaba-b-r1. Additionally, we examined other proteins that shape the biophysical response properties of neurons, such as the voltage-gated ion channels para and Ih.

The metabotropic GABA receptor subunit Gaba-b-r1 is the most highly expressed GABA receptor subunit in T4/T5 neurons after Rdl (*Pankova and Borst, 2016*; *Davis et al., 2020*). Gaba-b-r1 is one out of three G-protein-coupled GABA receptor subunits described in *Drosophila* and has been shown to be involved in sleep and appetitive long-term memory (*Mezler et al., 2001*; *Kim et al., 2017*; *Pavlowsky et al., 2018*). We inserted the FlpTag cassette in the MiMIC site between the first and second exon (MI01930) of the Gaba-b-r1 locus via RMCE. Again, two lines with two different orientations of the FlpTag cassette were obtained. The line with the cassette in the same orientation as the gene was used to observe the pan-neuronal distribution of the endogenous GABA receptor subunit. Gaba-b-r1 is expressed throughout all neuropils with strongest signal in the outer distal layers of the medulla and the medial part of the lobula (*Figure 6A*). Upon cell-type specific, FLP-dependent inversion of the FlpTag cassette in T4/T5 neurons, we could not observe any Gaba-b-r1::GFP signal in T4/T5 dendrites or axons (*Figure 6B*). Although RNAseq studies detected Gaba-b-r1 mRNA in T4/T5 neurons (*Pankova and Borst, 2016*; *Davis et al., 2020*), we could not confirm this result at the protein level.

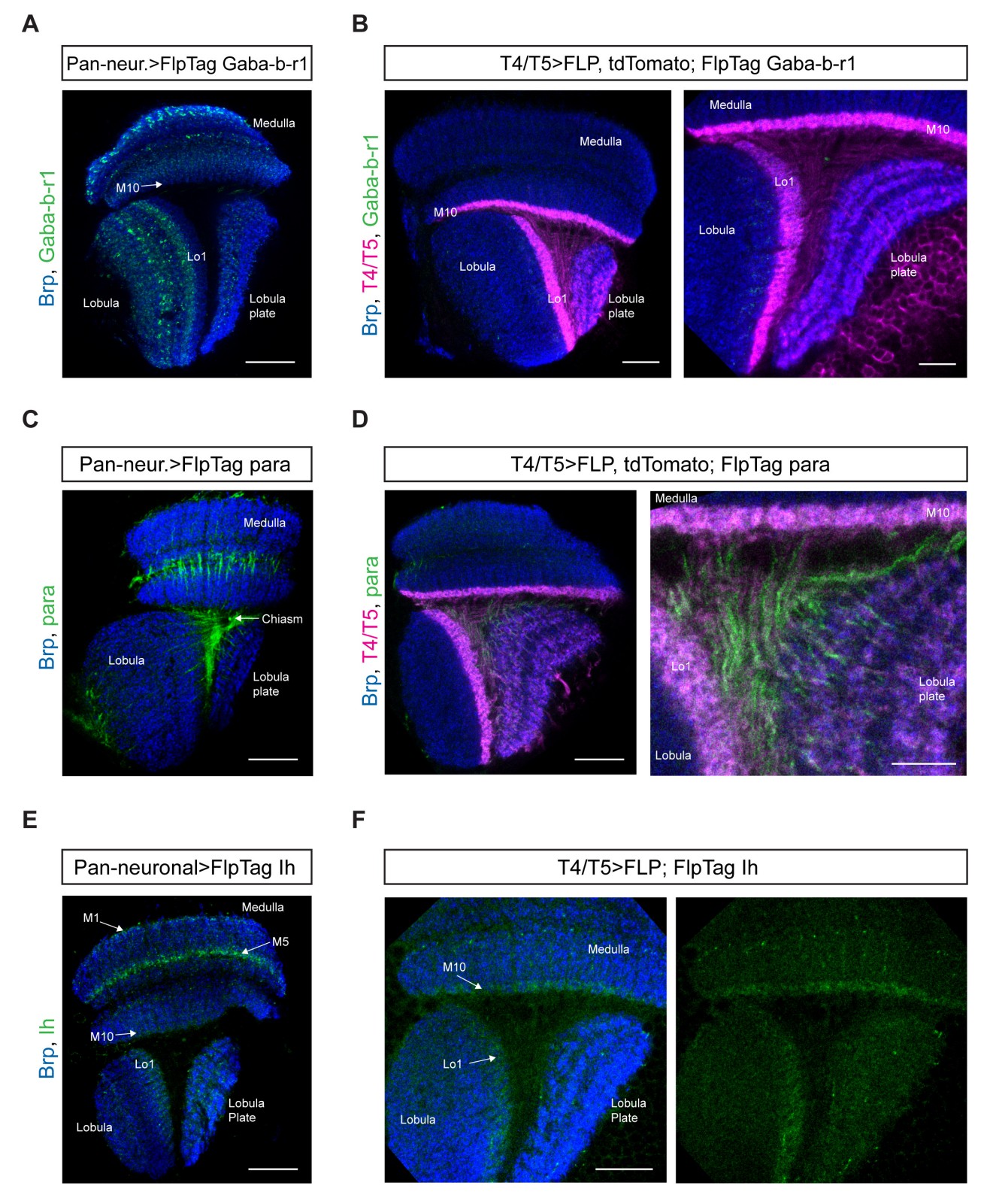

**Figure 6.** FlpTag lines for Gaba-b-r1, para and Ih. Optic lobes with pan-neuronal expression of FlpTag Gaba-b-r1 (**A**), FlpTag para (**C**), and FlpTag Ih (**E**). (**B**) Expression of FlpTag Gaba-b-r1 in T4/T5 neurons labeled with myr::tdTomato. Left panel: horizontal view on the optic lobe overview (scale bar: 20 μm). Right panel: close-up of medulla layer M10, lobula layer Lo1 and Lobula plate layers 1–4 (scale bar: 10 μm). (**D**) Expression of FlpTag para in T4/T5 neurons labeled with myr::tdTomato. Left panel: horizontal view on the optic lobe overview (scale bar: 20 μm). Right panel: close-up of medulla layer

*Figure 6 continued on next page*

*Figure 6 continued*

M10, lobula layer Lo1 and Lobula plate layers 1–4 (scale bar: 10 µm). (F) Expression of FlpTag Ih in T4/T5 neurons. Horizontal view on the optic lobe with medulla layer M10, lobula layer Lo1 and Lobula plate layers 1–4 (scale bar: 12 µm). Left panel: Background staining anti-brp in blue and. Right panel: Ih::GFP signal only.

Paralytic (para) is the only voltage-gated sodium channel described in *Drosophila* and highly expressed in T4/T5 neurons (*Pankova and Borst, 2016*). It is required for the generation of sodium-dependent action potentials. We created the FlpTag para line by inserting the FlpTag cassette into the MiMIC landing site between the first and second exon (MI08578), thereby covering all of its 60 isoforms. Surprisingly, the pan-neuronal expression pattern is rather sparse with some bundles labeled in the medulla across the serpentine layer and axonal fibers in the chiasm between medulla, lobula and lobula plate (*Figure 6C*). In the T4/T5 specific FlpTag genotype, para is strongly expressed in the axonal fibers connecting dendrites and axon terminals in T4/T5 neurons (*Figure 6D*).

Ih is a voltage-gated, hyperpolarization-activated ion channel which is highly expressed in T4/T5 neurons (*Chen and Wang, 2012*; *Hu et al., 2015*; *Pankova and Borst, 2016*). To generate the corresponding FlpTag line, the FlpTag cassette was inserted in the MiMIC site MI12136 housed by the coding intron between the first and second exons of the Ih gene locus. In the pan-neuronal FlpTag line, Ih is expressed most strongly in two layers of the distal medulla (M1 and M5), as well as in the lobula plate and in Lo1 of the lobula (*Figure 6E*). In the T4/T5-specific FlpTag genotype, Ih is localized to the T4 and T5 dendrite area in medulla layer 10 and lobula plate layer 1 (*Figure 6F*).

Taken together, we generated four working FlpTag lines which uncovered the differential subcellular distribution of the neurotransmitter receptor subunits GluClα and Gaba-b-r1 and the voltage-gated ion channels para and Ih. We demonstrated that the FlpTag approach is generalizable and can be expanded to many genes with MiMIC insertion sites.

## Discussion

Neurotransmitter receptors are essential neuronal elements that define the sign and temporal dynamics of synaptic connections. For our understanding of complex neural circuits, it is indispensable to examine which transmitter receptor types are used by the participating neurons and to which compartment they localize. Here, we developed FlpTag, a generalizable method for endogenous, cell-type-specific labeling of proteins. Alongside several GFP-tagged UAS-lines, we used our newly developed FlpTag lines to explore the distribution of receptor subunits GluClα, Rdl, Dα7, Gaba-b-r1 and voltage-gated ion channels para and Ih in motion-sensing T4/T5 neurons of the visual system of *Drosophila*. We found that these ion channels are localized to either the dendrite, the axonal fiber or the axon terminal (summarized in *Figure 7A and C*). Even at the level of individual dendrites, GluClα, Rdl and Dα7 were differentially distributed precisely matching the locations where T4 and T5 neurons sample signals from their glutamatergic, cholinergic, or GABAergic input neurons, respectively (summarized in *Figure 7*).

### Protein tagging methods: endogenous tags and UAS-lines

Working with *Drosophila* as model organism bears some unrivaled advantages when it comes to genetic tools. The MiMIC and FlyFos libraries, for instance, are large-scale approaches of enormous value for the fly community as they provide GFP-tagged protein lines for thousands of *Drosophila* genes including several neurotransmitter receptors and voltage-gated ion channels (*Venken et al., 2011*; *Nagarkar-Jaiswal et al., 2015a*; *Sarov et al., 2016*). Recently, Kondo et al. expanded these existing libraries with T2A-Gal4 insertions in 75 neurotransmitter receptor genes that can also be exchanged by the fluorescent protein tag Venus (*Kondo et al., 2020*). While all these approaches tag genes at their endogenous locus, none of them are conditional, for example they cannot be applied in a cell-type-specific manner. Hence, ascribing the expression of the pan-neuronally tagged proteins to cell-types of interest are challenging in dense neuronal tissue.

To overcome these difficulties, we used two conditional strategies for the investigation of membrane protein localizations in our cell types of interest, T4 and T5 neurons. First, we developed GFP-tagged *UAS*-lines for GluClα and Rdl and tested an existing *UAS-Dα7::GFP* line. As stated above,

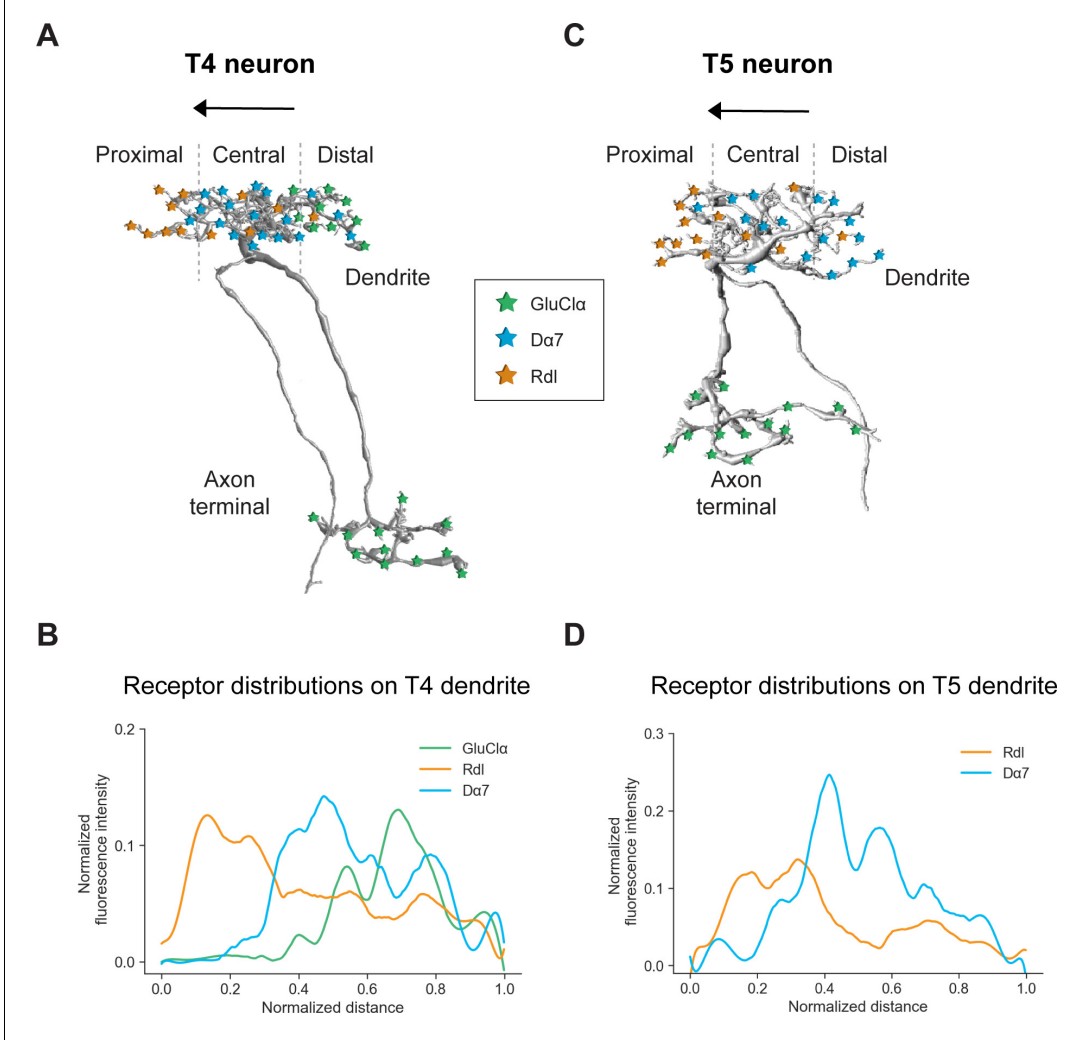

**Figure 7.** Summary of the receptor distributions of GluClα, Rdl and Dα7 in T4 and T5 neurons. (**A**) Scheme of EM-reconstructed T4 neuron with distribution of receptors on dendrite and axon terminal (image extracted from Seven medulla column connectome dataset, https://emdata.janelia.org/#/repo/medulla7column, #3b548, Janelia Research Campus). (**B**) Quantification of GluClα (green), Rdl (orange) and Dα7 (blue) distribution over the whole dendritic length (distance) averaged across several T4 from all subtypes (combined data from *Figures 4D* and *5D*). All dendrites were aligned pointing to the right with the most proximal point at 0.0 and the most distal point at 1.0. (**C**) Scheme of EM-reconstructed T5 neuron with distribution of receptors on dendrite and axon terminal (image extracted from Seven medulla column connectome dataset, https://emdata.janelia.org/#/repo/medulla7column, #3b548, Janelia Research Campus). (**D**) Rdl (orange) and Dα7 (blue) distribution over the whole dendritic length (normalized distance) averaged across several T5 from all subtypes (combined data from *Figures 3D* and *4D*). All dendrites were aligned pointing to the right with the most proximal point at 0.0 and the most distal point at 1.0.

aberrant localization of overexpressed proteins can occur, however, this is not always the case. Overexpression of *UAS-GluClα::GFP* shows a similar receptor localization pattern as both MiMIC and FlpTag endogenous lines (*Figure 2—figure supplement 1*), thus, validating the use of *UAS-GluClα:: GFP* for studying receptor distribution. Additionally, previous studies reported that the *UAS-Dα7:: GFP* line showed proper localization of the acetylcholine receptor to endogenous synapses when compared to antibody stainings or endogenous bruchpilot (Brp) puncta (*Kuehn and Duch, 2013*; *Mosca and Luo, 2014*). Here, we confirmed this finding and further showed that Dα7::GFP presumably localizes only to cholinergic synapses. Overexpressing Dα7::GFP in a medulla neuron that is devoid of endogenous Dα7 demonstrated that Dα7::GFP localized to apparent cholinergic synapses. Hence, the *UAS-Dα7::GFP* line can be used to study the distribution of cholinergic synapses, but not the exact composition of cholinergic receptor subunits. A recent study showed that quantitatively the levels of the postsynaptic density protein PSD95 change when overexpressed, but qualitatively

the localization is not altered (*Willems et al., 2020*). Altogether, this suggests that tagged overexpression lines can be used for studying protein localizations, but they have to be controlled carefully and drawn conclusions might be different for every line.

## The FlpTag method is generalizable and can be expanded to many genes

Ideally, a tool for protein tagging should be both endogenous and conditional. This can be achieved by introducing an FRT-flanked STOP cassette upstream of the gene of interest which was engineered with an epitope tag or fluorescent protein. Only upon cell-type specific expression of Flp, the tagged protein will be expressed in a cell-type specific manner. This genetic strategy was utilized by two independent studies to label the presynaptic protein Brp, the histamine channel ort and the vesicular acetylcholine transporter VAChT (*Chen et al., 2014*; *Pankova and Borst, 2017*). Recently, a new approach based on the split-GFP system was utilized for endogenous, conditional labeling of proteins in two independent studies (*Kondo et al., 2020*; *Luo et al., 2020*). However, all these aforementioned approaches are not readily generalizable and easily applicable to any gene of interest.

The FlpTag strategy presented here overcomes these caveats by allowing for endogenous, conditional tagging of proteins and by offering a generalizable toolbox for targeting many genes of interest. Similar to the conditional knock-out tools FlpStop and FlipFlop (*Fisher et al., 2017*; *Nagarkar-Jaiswal et al., 2017*), FlpTag utilizes a FLEx switch to conditionally control expression of a reporter gene, in our case GFP. Likewise, FlpTag also easily integrates using the readily available intronic MiMIC insertions. Here, we attempted to generate FlpTag lines for six genes, GluClα, Rdl, Dα7, Gaba-b-r1, para and Ih (overview of lines in *Table 1*). Four out of these six lines yielded conditional GFP-tagged protein lines (GluClα, Gaba-b-r1, para, Ih). We injected the FlpTag cassette in MI02620 for Rdl and MI12545 for Dα7, but could not observe any GFP expression across the brain (data not shown). The MiMiC insertion sites used for Rdl and Dα7 seem to be in a suboptimal location for tagging the protein.

## Expansion of the FlpTag toolbox

As of now, there are MiMIC insertions in coding introns for more than 2800 genes available, which covers approximately 24% of neuronal genes (*Venken et al., 2011*; *Nagarkar-Jaiswal et al., 2015a*; *Fisher et al., 2017*). Additionally, the attP insertion sites generated in the study by Kondo et al. provide possible landing sites for the FlpTag cassette for 75 neurotransmitter receptor genes (*Kondo et al., 2020*). Transmembrane proteins such as neurotransmitter receptors form complex 3D structures making fluorescent tagging especially difficult. Neither the MiMIC insertion sites, nor the target sites of the Kondo study at the C-terminus of several transmitter receptor genes, ensure a working GFP-tagged protein line. For genes of interest lacking a suitable MiMIC insertion site we generated a homology directed repair (HDR) cassette which utilizes CRISPR/Cas9-mediated gene

**Table 1.** Overview of available MiMIC GFSTF and FlpTag lines for investigated genes.

| | Gene | MiMIC insertion (coding intron) | MiMIC GFSTF existing | MiMIC GFSTF working | Chromosome | Phase | FlpTag working | Localization in T4/T5 neurons |
|---|---|---|---|---|---|---|---|---|
| 1 | GluClα | MI02890, MI14426 | MI02890 | Yes | III | 2 | Yes, MI02890 | T4: dendrites + terminals; T5: terminals |
| 2 | Rdl | MI02620, MI02957 | MI02620 | No | III | 0 | No, MI02620 | From UAS line: dendrites |
| 3 | Dα7 | MI12545 | This study (MI12545) | No | X | 1 | No | From UAS line: dendrites |
| 4 | Gaba-b-r1 | MI01930, MI05755 | MI01930 | Yes | II | 0 | Yes, MI01930 | No |
| 5 | para | MI08578 | This study (MI08578) | Yes | X | 0 | Yes, MI08578 | T4/T5 axonal fibers |
| 6 | Ih | MI03196, MI12136 | This study (MI12136) | Yes | II | 2 | Yes, MI12136 | T4/T5 dendrites |

editing to integrate the FlpTag cassette in any desired gene locus (*Supplementary file 6– 8*; *Gratz et al., 2014*; *Fisher et al., 2017*). The plasmid consists of the FlpTag cassette flanked by multiple cloning sites for the insertion of homology arms (HA). Through HDR the FlpTag cassette can be knocked-in into any desired locus. Taken together, the FlpTag cassette is a generalizable tool that can be integrated in any available attP-site in genes of interest (*Venken et al., 2011*; *Nagarkar- Jaiswal et al., 2015a*; *Kondo et al., 2020*) or inserted by CRISPR-HDR into genes lacking attP land- ing sites. This allows for the investigation of the endogenous spatial distributions of proteins, as well as the correct temporal dynamics of protein expression.

Further, the FlyFos project demonstrated that most fly lines with an extra copy of GFP-tagged protein-coding genes worked normally and GFP-tagged proteins could be imaged in living fly embryos and pupae (*Sarov et al., 2016*). In principle, live-imaging of the GFP-tagged lines we cre- ated could be performed during different developmental stages of the fruit fly. In general, the tools generated here can be used as specific postsynaptic markers, visualizing glutamatergic, GABAergic, and cholinergic synapses with standard confocal light microscopy. This extends the existing toolbox of *Drosophila* postsynaptic markers (*Sánchez-Soriano et al., 2005*; *Raghu et al., 2009*; *Andlauer et al., 2014*; *Chen et al., 2014*; *Petzoldt et al., 2014*; *Kondo et al., 2020*; *Luo et al., 2020*) for studying the localization and development of various types of synapses.

## Functional relevance of transmitter receptors and voltage-gated channels for *Drosophila* motion-sensitive neurons

T4/T5 neurons combine spatiotemporal input from their presynaptic partners, leading to selective responses to one of the four cardinal directions. Numerous studies investigated the mechanisms underlying direction-selective responses in T4/T5 neurons, yet the computation is still not fully understood. At an algorithmic level, a three-arm detector model is sufficient to describe how direc- tion-selective responses in T4/T5 neurons arise (*Arenz et al., 2017*; *Haag et al., 2017*). This model relies on the comparison of signals originating from three neighboring points in space via a delay- and-compare mechanism. The central arm provides fast excitation to the neuron. While one flanking arm amplifies the central signal for stimuli moving along the preferred direction, the other inhibits the central signal for stimuli moving along the null direction of the neuron. Exploring the neurotrans- mitter receptors and their distribution on T4/T5 dendrites allows us to define the sign as well as the temporal dynamics of some of the input synapses to T4/T5.

According to the algorithmic model, we expect an excitatory, amplifying input signal on the distal side of T4/T5 dendrites. Here, we found that T4 cells receive an inhibitory, glutamatergic input from Mi9 via GluClα, which, at first sight, seems to contradict our expectation. However, since Mi9 has an OFF-center receptive field (*Arenz et al., 2017*; *Richter et al., 2018*; *Drews et al., 2020*), this gluta- matergic synapse will invert the polarity from Mi9-OFF to T4-ON. Theoretically, in darkness, Mi9 inhibits T4 via glutamate and GluClα, and this inhibition is released upon an ON-edge moving into its receptive field. The concomitant closure of chloride channels and subsequent increased input resistance in T4 cells results in an amplification of a subsequent excitatory input signal from Mi1 and Tm3. As shown by a recent modeling study, this biophysical mechanism can indeed account for pre- ferred direction enhancement in T4 cells (*Borst, 2018*). Some studies failed to detect preferred direction enhancement in T4/T5 neurons and they proposed that the enhanced signal in PD seen in GCaMP recordings could be a result from a non-linear calcium-to-voltage transformation (*Gruntman et al., 2018*; *Gruntman et al., 2019*; *Wienecke et al., 2018*). If this was really the case, the role of Mi9 and GluClα must be reconsidered and future functional experiments will shed light onto this topic.

Nevertheless, Strother et al. showed that the RNAi- knock-down of GluClα in T4/T5 neurons leads to enhanced turning responses on the ball set-up for faster speeds of repeating ON and OFF edges (*Strother et al., 2017*). Although this observation cannot answer the question about preferred direc- tion enhancement in T4 cells, it indicates that both T4 and T5 receive inhibitory input and that removal of such create enhanced turning responses at the behavioral level. In line with these obser- vations, we also found the glutamate receptor GluClα in T4/T5 axon terminals. A possible functional role of these inhibitory receptors in the axon terminals could be a cross-inhibition of T4/T5 cells with opposite preferred directions via lobula plate intrinsic neurons (LPis). Glutamatergic LPi neurons are known to receive a cholinergic, excitatory signal from T4/T5 neurons within one layer and to inhibit lobula plate tangential cells, the downstream postsynaptic partners of T4/T5 neurons, via GluClα in

the adjacent oppositely tuned layer. This mechanism induces a motion opponent response in lobula plate tangential cells and increases their flow-field selectivity (*Mauss et al., 2015*). In addition, LPi neurons could also inhibit T4/T5 neurons presynaptically at their axon terminals via GluClα in order to further sharpen the flow-field selectivity of lobula plate tangential cells. Taken together, exploring the subcellular distribution of GluClα in T4/T5 neurons highlights its differential functional roles in different parts of these cell types.

Secondly, the Dα7 signal in the center of T4/T5 dendrites discovered here, corresponds to ionotropic, cholinergic input from Mi1 and Tm3 for T4, and Tm1, Tm2 and Tm4 for T5. These signals correspond to the central, fast, excitatory arm of the motion detector model. As T4 and T5 express a variety of different ACh receptor subunits (*Davis et al., 2020*), the exact subunit composition and underlying biophysics of every cholinergic synapse on T4/T5 dendrites still awaits further investigations.

Third, inhibition via GABA plays an essential role in creating direction-selective responses in both T4 and T5 neurons (*Fisher et al., 2015a*; *Arenz et al., 2017*; *Strother et al., 2017*; *Gruntman et al., 2018*) by providing null direction suppression. Computer simulations showed that direction selectivity decreases in T4/T5 motion detector models without this inhibitory input on the null side of the dendrite (*Arenz et al., 2017*; *Borst, 2018*; *Strother et al., 2017*). Here, we show that T4 and T5 neurons possess the inhibitory GABA receptor subunit Rdl mainly on the proximal base on the null side of their dendrites, providing the synaptic basis for null direction suppression. We did not detect the metabotropic GABA receptor subunit Gaba-b-r1 in T4/T5 neurons using the newly generated FlpTag Gaba-b-r1 line. Finally, all of the receptor subunits GluClα, Rdl and Dα7 investigated here are ionotropic, fast receptors, which presumably do not add a temporal delay at the synaptic level. In the detector model described above, the two outer arms provide a slow and sustained signal, and such properties are already intrinsic properties of these input neurons (*Arenz et al., 2017*; *Serbe et al., 2016*). However, we cannot exclude that slow, metabotropic receptor subunits for acetylcholine or GABA (e.g. Gaba-br2) which are also present in T4/T5 and could induce additional delays at the synaptic level (*Takemura et al., 2011*; *Davis et al., 2020*).

Furthermore, we investigated the subcellular distribution of the voltage-gated ion channels para and Ih in T4/T5 neurons. We found para, a voltage-gated sodium channel, to be distributed along the axonal fibers of both T4 and T5 neurons. As para is important for the generation of sodium-dependent action potentials, it will be interesting for future functional studies to investigate, if T4/T5 really fire action potentials and how this shapes their direction-selective response. Further, we detected Ih, a voltage-gated ion channel permeable for several types of ions, in T4/T5 dendrites using the FlpTag strategy. Ih channels are activated at negative potentials below −50 mV and as they are permeable to sodium and potassium ions, they can cause a depolarization of the cell after hyperpolarization (*Magee, 1999*; *Littleton and Ganetzky, 2000*; *George et al., 2009*). Loss-of-function studies will unravel the functional role of the Ih channel for direction-selective responses in T4/T5 neurons.

## Outlook

Since the ability to combine synaptic inputs from different neurotransmitters at different spatial sites is common to all neurons, the approaches described here represent an important future perspective for other circuits. Our tools can be used to study the ion channels GluClα, Rdl, Dα7, Gaba-b-r1, para and Ih in any given *Drosophila* cell-type and circuit. Furthermore, the FlpTag tool box can be used to target many genes of interest and thereby foster molecular questions across fields.

The techniques described here can be transferred to other model organisms as well, to study the distribution of different transmitter receptors. For instance, in the mouse retina - similar to motion-sensing T4/T5 neurons in the fruit fly - so-called On-Off direction-selective ganglion cells receive asymmetric inhibitory GABAergic inputs from presynaptic starburst amacrine cells during null-direction motion. A previous study investigated the spatial distribution of GABA receptors of these direction-selective ganglion cells using super-resolution imaging and antibody staining (*Sigal et al., 2015*). Additionally, starburst amacrine cells also release ACh onto ganglion cells which contributes to the direction-selective responses of ganglion cells. Thus, mapping the distribution of ACh receptors on direction-selective ganglion cells will be the next important step to further investigate cholinergic transmission in this network (*Sethuramanujam et al., 2020*).

Overall, we demonstrated the importance of exploring the distributions of neurotransmitter receptors and ion channels for systems neuroscience. The distinct distributions in T4/T5 neurons discovered here and the resulting functional consequences expand our knowledge of the molecular basis of motion vision. Although powerful, recent RNAseq studies lacked information about spatial distributions of transmitter receptors which can change the whole logic of wiring patterns and underlying synaptic signs. Future studies can use this knowledge to target these receptors and directly probe their role in functional experiments or incorporate the gained insights into model simulations. However, this study is only highlighting some examples of important neural circuit components: expanding the approaches described here to other transmitter receptors and ion channels, as well as gap junction proteins will reveal the full inventory and the spatial distributions of these decisive determinants of neural function within an individual neuron.

## Materials and methods

### Fly strains
Flies were raised at 25°C and 60% humidity on standard cornmeal agar medium at 12 hr light/dark cycle. The following driver lines were used: *R42F06-Gal4* to label T4/T5 neurons, *R57C10-Gal4* for addressing all neurons, *SS03734-splitGal4* to address L1, *R19F01-AD; R71D01-DBD* to address Mi1, *10–50* Gal4 to label T1, and *Dα7-TG4* (BL#77828). The *T4-splitGal4* line was generated by combining the hemidriver lines *VT16255-AD* (BL#75205) and *VT12314-DBD* (unpublished, T. Schilling); the *T5-splitGal4* line was generated by combining the hemidriver lines *VT13975-AD* and *R42F06-DBD* (unpublished, T. Schilling). The following *UAS*-reporter lines were used for labeling cell-types and drive flippase-expression: *UAS-myr::tdTomato* (BL#32222), and *UAS-FLP1.D* (BL#4539). For labeling individual T4/T5 neurons stochastically together with the receptor lines, we combined *UAS-myr:: tdTomato; UAS-GluClα::GFP/UAS-Rdl::GFP/UAS-Dα7::GFP* with *hs-FLP; FRT-Gal80-FRT; R42F06-Gal4* and heat-shocked pupae (P1-P3) for 5–8 min at 37°C in a water bath.

### Generation of new genetic *UAS*-lines
The coding sequencing (CDS) of *GluClα* isoform K was acquired from flybase.org and along with the sequence of *GFP* flanked by 4xGGS linker was synthesized by Eurofins Genomics and inserted into pEX-A258 backbone between NotI and XbaI restriction sites. Using restriction digestion with NotI and XbaI the *GluClα* fragment was cloned into *pJFRC7-20XUAS-IVS-mCD8::GFP* (*Pfeiffer et al., 2010*) vector. Similarly, the CDS of *Rdl* isoform F was acquired from flybase.org and with the sequence of *GFP* flanked by 4xGGS linker was synthesized as three DNA fragments by Invitrogen GeneArt Gene Synthesis. Each fragment carried a complementary overlapping section of 25–35 bps on both ends. *pJFRC7-20XUAS-IVS-mCD8::GFP* (*Pfeiffer et al., 2010*) vector was digested with NotI and XbaI restriction enzymes and all three DNA fragments were inserted using NEBuilder HiFi DNA Assembly. Embryo injections were performed by BestGene Inc (Chino Hills, CA, USA).

For the generation of the FlpTag constructs, the pFlip-Flop-P0 plasmid (*Nagarkar-Jaiswal et al., 2017*) ordered from *Drosophila* Genomics Resource Center (NIH Grant 2P40OD010949) was digested with BsmFI and EcoRI leaving the plasmid backbone with FRT, FRT14 and attB sites. Six DNA fragments were synthesized by Invitrogen GeneArt Gene Synthesis. Three fragments contained a predicted splice donor site (one for each phase) and half of an inverted 4xGGS-GFP. The other three contained half of an inverted GFP-4xGGS followed by a slice acceptor (SA) site (one for each phase). All fragments had complementary overlapping sections of 25–35 bps which was used to insert phase-paired fragments into the digested pFlip-Flop plasmid using NEBuilder HiFi DNA Assembly. Embryo injections were performed by BestGene Inc (Chino Hills, CA, USA), including PCR-verifications and balancing.

### S2 Schneider cell culture
We used *Drosophila* S2R+ Schneider cells in culture *Drosophila* Genomics Resource Center, stock #150 for testing the newly generated *UAS*-receptor::GFP constructs before embryo injections. S2R+ cells were cultured in Schneider's *Drosophila* medium (Thermo Fisher Scientific) supplemented with 10% fetal bovine serum (Thermo Fisher Scientific) and penicillin/streptomycin (Cytiva). *UAS*-constructs were tested by transfecting 250 ng of *UAS*-plasmid and 250 ng of *actin5C-Gal4* plasmid (gift

from T. Kornberg) in 24-well plates using the FuGENE HD Kit (Promega). Two days later, we checked for GFP-expression in transfected S2 cells with a fluorescence binocular microscope.

## Immunohistochemistry

Fly brains were dissected in cold 0.3% PBST and fixed in 4% PFA in 0.3% PBST for 25 min at room temperature. Subsequently, brains were washed four to five times in 0.3% PBST and blocked in 10% normal goat serum (NGS) in 0.3% PBST for 1 hr at room temperature. Primary antibodies used were mouse anti-Bruchpilot Brp (nc82, Developmental Studies Hybridoma Bank, 1:20, RRID:AB_2314867), rabbit anti-dsRed (Takara Bio, 1:300, RRID:AB_10013483), and rat anti-Dα7 (gift from H. Bellen, 1:2000). Secondary antibodies used were: goat anti-mouse ATTO 647N (Rockland, 1:300, RRID:AB_2614870), goat anti-rabbit Alexa Fluor 568 (Thermo Fisher Scientific, 1:300, RRID:AB_10563601), and goat anti-rat Alexa Fluor 647 (Thermo Fisher Scientific, 1:300, RRID:AB_141778). GFP-labeled receptors were imaged natively without antibody staining. 5% NGS was added to all antibody solutions and both primary and secondary antibodies were incubated for at least 48 hr at 4°C. Brains were mounted in Vectashield Antifade Mounting Medium (Vector Laboratories) and imaged on a Leica TCS SP8 confocal microscope equipped with 488-, 561-, and 633 nm lasers, using a 63X glycerol objective.

## Quantifications of receptor distributions and number of puncta

For intensity quantification, confocal stacks were processed in ImageJ using maximum intensity projection. These images were then analyzed in python using the Skimage and Numpy packages. For each image, florescence was normalized to the maximum intensity within an image. Additionally, images were cropped to include the entire dendritic cross section and aligned pointing to the right with the most proximal point to the left and the most distal point to the right. These images were normalized to the maximum cropped image length.

For quantification of number of receptor puncta, confocal stacks were taken from the entire cross-section of the dendrite as above. Puncta were counted in ImageJ software using the 3D object counter plugin of Fiji (*Bolte and Cordelières, 2006*).

## Statistical analysis

Statistical significance was tested with a Student t-test when comparing two groups. A p-value below 0.05 was considered significant. In the case of pan-neuronal quantification where multiple groups were compared, statistical significance was tested using one-way ANOVA. In all figures, * was used to indicate a p-value<0.05, ** for p<0.01, and *** for p<0.001. Statistical analysis and graphs were generated in Python 3.4 using SciPy and Seaborn packages respectively. Figures were generated in Adobe Illustrator CC.

## Acknowledgements

We are grateful to T Schilling for unpublished driver lines, the Bloomington *Drosophila* Stock Center for fly lines and the *Drosophila* Genomics Resource Center (supported by NIH grant 2P40OD010949) for S2 cells; T Kornberg (UCSF) for the *actin-Gal4* plasmid for S2 cell culture and H Bellen for the Dα7-antibody. We thank W Essbauer for technical assistance with molecular work and R Kutlesa and I Ribeiro for help with the S2 cell culture; T Schilling and J Pujol-Marti for discussions and support throughout all stages of this project; I Ribeiro, A Mauss, L Groschner, G Ammer, J Malis, A Harbauer and A Barker for carefully reading the manuscript; M Drews and N Pirogova for helping with programming.

## Additional information

### Competing interests

Alexander Borst: Reviewing editor, *eLife*. The other authors declare that no competing interests exist.

## Funding

| Funder | Author |
|--------|--------|
| Max-Planck-Gesellschaft | Sandra Fendl<br>Renee Marie Vieira<br>Alexander Borst |

The funders had no role in study design, data collection and interpretation, or the decision to submit the work for publication.

## Author contributions

Sandra Fendl, Conceptualization, Formal analysis, Validation, Investigation, Visualization, Methodology, Writing - original draft, Project administration, Writing - review and editing, Conceived and designed the study, Imaged all data shown and processed confocal images, Wrote the manuscript and prepared the figures with the help of RMV and AB; Renee Marie Vieira, Conceptualization, Software, Formal analysis, Visualization, Methodology, Writing - review and editing, Conceived and designed the study; Analyzed and quantified all imaged data; Developed and created the UAS-lines and the FlpTag-construct and stocks with the help of SF; Alexander Borst, Conceptualization, Resources, Supervision, Project administration, Writing - review and editing

## Author ORCIDs

Sandra Fendl ⓘ https://orcid.org/0000-0001-6442-2542
Renee Marie Vieira ⓘ https://orcid.org/0000-0001-8520-7382

## Decision letter and Author response

Decision letter https://doi.org/10.7554/eLife.62953.sa1
Author response https://doi.org/10.7554/eLife.62953.sa2

# Additional files

## Supplementary files

- Supplementary file 1. Plasmid map of the full sequence of pJFRC7-20xUAS-GluClα-GFP.
- Supplementary file 2. Plasmid map of the full sequence of pJFRC7-20xUAS-Rdl-GFP.
- Supplementary file 3. Plasmid map of the full sequence of pUC57-FlpTag-GFP-ph0.
- Supplementary file 4. Plasmid map of the full sequence of pUC57-FlpTag-GFP-ph1.
- Supplementary file 5. Plasmid map of the full sequence of pUC57-FlpTag-GFP-ph2.
- Supplementary file 6. Plasmid map of the full sequence of pHD-FlpTag-DsRed-HDR-ph0.
- Supplementary file 7. Plasmid map of the full sequence of pHD-FlpTag-DsRed-HDR–ph1.
- Supplementary file 8. Plasmid map of the full sequence of pHD-FlpTag-DsRed-HDR-ph2.
- Supplementary file 9. 3D-image of a T4 dendrite (subtype d) (magenta) with GluClα::GFP (green).
- Supplementary file 10. 3D-image of a T4 dendrite (subtype d) (magenta) with Rdl::GFP (yellow).
- Supplementary file 11. 3D-image of a T4 dendrite (subtype d) (magenta) with Dα7::GFP (cyan).
- Transparent reporting form

## Data availability

All data generated or analysed during this study are included in the manuscript and supporting files. Source data files have been provided for Figures 2, 3 and 4. Instructions on accessing the seven medulla column connectome data are available at https://github.com/janelia-flyem/Connectome-Hackathon2016/wiki/Accessing%20Optic%20Lobe%20Dataset%20using%20Google%20Cloud.

The following previously published dataset was used:

| Author(s) | Year | Dataset title | Dataset URL | Database and Identifier |
|-----------|------|---------------|-------------|-------------------------|
| Shinomiya K | 2019 | Seven medulla column connectome | http://emdata.janelia. | https://emdata. |

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
