## [Decision Letter]

**Acceptance summary:**

This paper describes tools that will be useful for studying the subcellular localization of neurotransmitter receptors. GluCla should also be a useful tool as well as the GABAR, para, and Ih FlpTag alleles. The work is carefully executed work and the manuscript is clearly written. We still hope that, in the future, you will be able to expand and generalize the use of the FlpTag approach. The manuscript also provides an interesting description of the localization of Glutamatergic, Cholinergic and GABAergic synapses in the fly visual system.

**Decision letter after peer review:**

[Editors’ note: the authors submitted for reconsideration following the decision after peer review. What follows is the decision letter after the first round of review.]

Thank you for submitting your work entitled "Highly specific subcellular distribution of neurotransmitter receptors in motion-sensing neurons" for consideration by *eLife*. Your article has been reviewed by four peer reviewers, and the evaluation has been overseen by a Reviewing Editor and a Senior Editor. The following individuals involved in review of your submission have agreed to reveal their identity: Filipe Pinto Teixeira (Reviewer #2); Tom Baden (Reviewer #3).

Our decision has been reached after extensive consultation between the reviewers. Based on these discussions and the individual reviews below, we regret to inform you that your work will not be considered further for publication in *eLife*.

The reviewers agree that the work is of quality but feel that the advances, both in terms of receptor localization in the T4/T5 circuit, and technical with the generation of new reporter lines, are incremental and would not be sufficient to be published in *eLife*.

As you will see below, the reviewers feel that the visualization of the localization of a small set of neurotransmitter receptors is in line with previous findings from TEM data while no further functional notions have been gained. The reviewers found it important to validate the role of the receptors in shaping the processing of the T4/T5 signal, either using functional imaging or electrophysiology/pharmacology.

The reviewers would have found it very interesting if you could have demonstrated what you suggest, that the localization of neurotransmitter receptors is intrinsic rather than induced by the presence of presynaptic sites. You should rephrase your claim that neurotransmitter receptor localization is intrinsic since you do not provide any evidence for that, which could be tested by manipulating these pre-synaptic sites that would have allowed you to test this important notion.

The reviewers also considered that the technical advances are limited: the UAS reporters are useful for the specific point that you address, and very well controlled, but they do not represent important tools that go beyond what is already available. Of course, the Flptag method could be such advance, but it is only described superficially and with little information on its efficiency and the possibility to develop it as a general method. It has only been used for one line, it relies on MiMic insertions and has not yet been tested with Crispr/Cas9. Furthermore, although the Flptag is a useful tool, it does not allow the tagging of individual neurons and thus it does not offer the fundamental and general tool needed to investigate localization at the single cell level in other systems. Similar methods with a larger coverage have already been published.

In conclusion, the part of the paper about receptor localization and how this helps refine our understanding of the T4T5 computation, needs validation of the role of the receptors in the shape of T4/T5 signal processing. In addition, you need to show the Flptag is a general method by targeting other genes and validating the Crispr/Cas9, and allowing the marking of single cells.

Reviewer #1:

To gain insights into the organization and function of neural circuits, knowledge about the distribution of synapses, neurotransmitters released on the pre-synaptic sides and neurotransmitter receptors on the postsynaptic side is invaluable. In this manuscript, S. Fendl, R. Vieira and A. Borst present genetic approaches to visualize the distinct subcellular distribution of three neurotransmitter receptors in motion-direction selective neurons T4 and T5 in the *Drosophila* visual system. The genetic tools include over-expression of UAS transgenes for receptor units GluClalpha (Glutamate), Rdl (GABA) and Dalpha7 (ACh). Furthermore, a novel approach named FlpTag is presented, that allows the labeling of GluClalpha by inserting an invertible GFP tag in the endogenous locus. In line with the known differential distribution of presynaptic sites along the dendrites of T4 and T5 neuron subtypes, the authors report matching receptor unit distributions. The observed patterns are extensively discussed within the context of the recent model for computations underlying motion-direction selectivity by the Borst laboratory. The study is descriptive, and as it assesses only 3 receptor subunits, also not comprehensive, this providing limited new insights. The FlpTag method is elegant, but as it is employed only for one receptor unit, and efficiencies in generating lines with this method are not provided, it remains uncertain, whether it is an easily implementable tool together for MiMIC lines or proposed novel Crispr/Cas9 mediated integration.

Specific comments:

1) The authors describe that GluClalpha is expressed in a matching pattern to what is expected from EM studies and known identities of pre-synaptic neurons. The patterns are similar when assessing over-expression and endogenous labeling techniques. The authors describe this must be an intrinsic property. But how can they conclude this? A novel insight would have been to use these new tools to address this point. For instance, by altering the pre-synaptic innervation, the assessment of the distribution of these receptors can be assessed. If it is unaltered, then the notion that distribution is intrinsic would gain support. An important aspect would also be to show the matching of pre-and postsynaptic markers to provide formal evidence that the puncta indeed correspond to synapses, and that matching occurs (in vertebrate synapses receptors are detected at the synapse, but also preisynaptically.)

2) The authors generate one line using their new FlpTag tool and conclude that this approach is generalizable. However, while it is likely on paper to be generalizable, the method would need to be described in detail regarding its efficiency with this and a few more factors. Alternatively, the significance of the method would need to be described with more careful wording.

3) "Upon cell-type specific expression of flippase recombinase (Flp), an intronic GFP-label is inverted and stably integrated into the gene as an artificial exon". And "Upon ΦC31-dependent integration of the FlpTag cassette into a coding intron of the GluCla target gene, the cassette is inverted, and spliced out together with the intron and no GFP-labeling occurs." Neither description seems quite right, phiC31 dependent-integration is not responsible for the inversion. Cassette exchange can be in either direction, but during injection, the inverted orientation is selected for. Also, Flp is not responsible for the integration, just the inversion. The method should be described correctly and in more detail with numbers allowing users to assess efficiency both to generate the lines and regarding inversion events. The line has been tested with Gal4 UAS-FLP for mediating inversion, which provides cell-type specificity, but it is not clear how efficient the method is at the single cell level with heat-shock during pupal development. Furthermore, for the other heat shock involving experiments in this study, it would be important to provide more details, including when and how many heat shock treatments are provided.

4) For the ACh receptor Dalpha7, the authors showed that GFP labeled puncta localized correctly, but that the numbers were less than expected from what is known about cholinergic input. It is proposed that another subunit/type of receptor may be important in this context, but it is not clear, why this was not further pursued.

5) The authors also assess the distribution of receptors in axons. The Discussion is extensive, highlighting, that "the subcellular distribution… highlights its differential functional roles in different parts of these cell types". While, it is clear that the receptors are indicative of different connections, the experiments shown in this study do not provide any functional insights, in particular how inhibitory input at the dendritic level and again at the axonal level could be integrated during information processing of T4/T5 neurons.

6) Subsection “Methods for investigation of receptor distributions”, the manuscript discusses the possibility of using their construct also for Crispr/Cas9 mediated insertion by a HDR cassette, referring to supplementary files solely with sequences. This is an important aspect and would benefit from a more detailed presentation and evidence for functionality to support the claim and the technical emphasis of the study in general.

Reviewer #2:

The tools the authors developed expand the repertoire of similar reagents already available to access neurotransmitter receptor expression in the fly. Additionally, the novel FlpTag tool allows to endogenously tag any gene of interest in a cell type specific manner expanding its usefulness to different molecular questions in various fields. The paper is well written, the results are solid and well supported but fall short. The results are limited to the validation of a previously published *UAS-Da7::GFP* line, two UAS lines the authors generated (*UAS-GluCla::GFP* and *UAS-Rdl::GFP*), the novel FlpTag tool, and to the description of the studied neurotransmitter receptors expression in the T4/T5 circuit. More specifically, the authors show that the GluCla, Rdl and Da7 distribution in T4 and T5 dendrites matches the known EM-reconstructed synapse distribution and neurotransmitter identity of upstream input neurons and it is compatible with one of several algorithmic models that describe how motion direction-selective responses in T4/T5 neurons is established. They further show a previously unknown expression of the glutamate receptor GluCla in T4/T5 axon terminals and discuss its possible role. As such, I find the manuscript in its current form better suited for a Tools and Resources article. I would support such publication.

Reviewer #3:

The paper uses a combination of novel genetic and classical immunohistochemical tools to pinpoint the receptor identities and their spatial distributions on T4/5 cells in the fruit fly visual system. In general, spatially offset receptor distributions here are known key ingredients to motion computation and understanding them in depth is certainly a useful thing to do. The paper extends previous results based on transcriptomics and EM (and other approaches) to lay out very nicely which receptors sit in which part of the dendrites. Overall, the results are very much in line with previous work and present an important consolidation of the literature. The paper is generally well written and quite clear. Accordingly, I support publication in *eLife* in principle.

However, it may be useful for the authors to consider some of the below

1) Stats + quantification: most statistics (all except Figure 2—figure supplement 1D) in the manuscript are summarised by bar plots with an error bar. This could do with an upgrade. E.g. convert to box/whisker-plots (or similar) and add raw data as a scatter in the background. Also, I notice the authors used t-tests in a few places, but I cannot see a test for normal distribution in the data (and since the raw data is not shown I also cannot eyeball this).

2) Related, the histograms (e.g. 2E, 4D, 5D, 6B,D) all seem to represent a mean across a bunch of experiments. These need some form of quantification of variance. A simple option could be confidence interval shadings. Moreover, they could do with stats. e.g. when comparing 2 distributions, one could fit a Generalized Additive Model (GAM) to pinpoint (i) if and (ii) where they are different (or something similarly telling).

Related to the above, I would have expected the receptor distributions to be quantified over 2D dendritic area, rather than "just" the linear distance along the dendrite's major axis. I appreciate that this is the axis that matters for the computation, but since the data is available in 2D, why not use it? Maybe there is additional nuance to be gained? The authors could compute the mean density map of receptors and plot it next to their example images. This would presumably also aid to more intuitively highlight the spatial offsets? It might make for a nice summary figure as well.

Reviewer #4:

Fendl et al. develop new labeling strategies to assess the asymmetric distribution of neurotransmitter subunits for glutamate, GABA, and Ach in the visual system. In particular, they develop ways to visualize GluClα, Dα7, and Rdl in motion-sensing T4/T5 neurons in *Drosophila*. The authors do a great job of carefully controlling for possible artifacts of overexpression in the channels they overexpress and show compelling evidence of distinct localization patterns of these three receptors in dendrites of the T4/T5 neurons. Overall, the manuscript is well written and presents both technical and biological advances, although both are somewhat incremental in significance. On the technical side, the development of the "FlpTag" tool for conditional labeling of proteins uses well-established technology and similar approaches have already been utilized. In addition, only a single receptor, GluClα, was actually successfully targeted for this FlpTag labeling. Beyond this technical achievement, the authors essentially confirm the localization of the three neurotransmitter receptors expected by the distribution of EM-reconstructed synapses of the relevant input neurons in the visual system. Thus, the contribution here of generating UAS-receptor-tags and the GluClα FlpTag to confirm putative receptor distributions appears to be an incremental contribution to the field. That being said, there are some additions that can improve the impact and rigor of this study.

1) Technical development: The authors used their FlpTag approach on GluClα and did not establish a toolkit targeting other neurotransmitter receptor subtypes. It's not clear why this strategy wasn't used for other receptors, at least for the two other receptors characterized in this study (ACh Dα7 and Rdl). Furthermore, a recent publication in *eLife* demonstrated that GluClα can be tagged conditionally as well (Molina-Obando et al., 2019), so it's not clear how important this reagent will prove to be to the field by targeting just this receptor. An expanded toolkit, akin to the one tagging all neurotransmitter receptors in the fly genome at their endogenous locus using T2A-Gal4 knock-in (Kondo et al., 2020), would be a great addition for the field but it is understood this is beyond the scope of the investigators interest in this study.

2) Biological insights: It is commendable the extent the authors went to carefully control for possible artifacts of mis-localization of the three neurotransmitter receptors due to overexpression of the straightforward UAS-receptor-tags. Beyond overexpression, it is always possible that the tags themselves may alter receptor trafficking, biochemical interactions, or integration into functional receptors. Can the authors comment in more detail on controls used to determine the tags do not impact receptor localization or function?

At least for the three receptors examined in this study, overexpression doesn't appear to lead to mis-localization or apparent artifacts. However, in terms of the biological significance, it's not clear how much is learned. Essentially, all the receptors mark synapses in areas they were suspected to be based on the neuronal inputs, and functional imaging or electrophysiology/pharmacology would be necessary to validate the ideas of how these receptors and their presumed physiological impacts actually shape signal processing in the T4/T5 neurons.

3) Although it is important to define which receptor subtypes exist at particular synapses, the experiments presented here by the authors do not really establish the receptors that function at the dendrites in the T4/T5 neurons. First, as the authors point out, it is a mystery what Ach receptor subtype(s) actually exist and function at dendrites in this system, and even whether Dα7 is playing a functional role. Second, while GluClα appears to be present at glutamatergic synapses, can the authors rule out the possibility of excitatory glutamate receptors also being present in T4/T5 neurons? It seems plausible that some of the kainate, AMPA, and/or NMDA type glutamate receptors encoded in the fly genome could also exist in this system, and if so these would have major impacts on the assumptions of signal processing proposed by the authors. Third, and for similar reasons, the findings on GABA receptors may be limited. Finally, also as pointed out by the authors, there are metabotropic receptors that could be expressed and play important roles that remain at present unknown.

---

## [Author Response]

[Editors’ note: the authors resubmitted a revised version of the paper for consideration. What follows is the authors’ response to the first round of review.]

Reviewer #1:To gain insights into the organization and function of neural circuits, knowledge about the distribution of synapses, neurotransmitters released on the pre-synaptic sides and neurotransmitter receptors on the postsynaptic side is invaluable. In this manuscript, S. Fendl, R. Vieira and A. Borst present genetic approaches to visualize the distinct subcellular distribution of three neurotransmitter receptors in motion-direction selective neurons T4 and T5 in the *Drosophila* visual system. The genetic tools include over-expression of UAS transgenes for receptor units GluClalpha (Glutamate), Rdl (GABA) and Dalpha7 (ACh). Furthermore, a novel approach named FlpTag is presented, that allows the labeling of GluClalpha by inserting an invertible GFP tag in the endogenous locus. In line with the known differential distribution of presynaptic sites along the dendrites of T4 and T5 neuron subtypes, the authors report matching receptor unit distributions. The observed patterns are extensively discussed within the context of the recent model for computations underlying motion-direction selectivity by the Borst laboratory. The study is descriptive, and as it assesses only 3 receptor subunits, also not comprehensive, this providing limited new insights. The FlpTag method is elegant, but as it is employed only for one receptor unit, and efficiencies in generating lines with this method are not provided, it remains uncertain, whether it is an easily implementable tool together for MiMIC lines or proposed novel Crispr/Cas9 mediated integration.Specific comments:1) The authors describe that GluClalpha is expressed in a matching pattern to what is expected from EM studies and known identities of pre-synaptic neurons. The patterns are similar when assessing over-expression and endogenous labeling techniques. The authors describe this must be an intrinsic property. But how can they conclude this? A novel insight would have been to use these new tools to address this point. For instance, by altering the pre-synaptic innervation, the assessment of the distribution of these receptors can be assessed. If it is unaltered, then the notion that distribution is intrinsic would gain support. An important aspect would also be to show the matching of pre-and postsynaptic markers to provide formal evidence that the puncta indeed correspond to synapses, and that matching occurs (in vertebrate synapses receptors are detected at the synapse, but also preisynaptically.)

We agree that we are not providing any evidence for the conclusion about intrinsic mechanisms. Hence, we removed this sentence from the text.

2) The authors generate one line using their new FlpTag tool and conclude that this approach is generalizable. However, while it is likely on paper to be generalizable, the method would need to be described in detail regarding its efficiency with this and a few more factors. Alternatively, the significance of the method would need to be described with more careful wording.

We changed the manuscript to a Resources and Tools format and included several new FlpTag lines.

3) "Upon cell-type specific expression of flippase recombinase (Flp), an intronic GFP-label is inverted and stably integrated into the gene as an artificial exon". And "Upon ΦC31-dependent integration of the FlpTag cassette into a coding intron of the GluCla target gene, the cassette is inverted, and spliced out together with the intron and no GFP-labeling occurs." Neither description seems quite right, phiC31 dependent-integration is not responsible for the inversion. Cassette exchange can be in either direction, but during injection, the inverted orientation is selected for. Also, Flp is not responsible for the integration, just the inversion. The method should be described correctly and in more detail with numbers allowing users to assess efficiency both to generate the lines and regarding inversion events. The line has been tested with Gal4 UAS-FLP for mediating inversion, which provides cell-type specificity, but it is not clear how efficient the method is at the single cell level with heat-shock during pupal development. Furthermore, for the other heat shock involving experiments in this study, it would be important to provide more details, including when and how many heat shock treatments are provided.

We thank the reviewer for his/her careful reading. We changed the respective sentences in the figure legends. For more details about the generated lines we added Table 1. Information about the heat shock involving experiments can be found in the Materials and methods section.

4) For the ACh receptor Dalpha7, the authors showed that GFP labeled puncta localized correctly, but that the numbers were less than expected from what is known about cholinergic input. It is proposed that another subunit/type of receptor may be important in this context, but it is not clear, why this was not further pursued.

This was not further pursued since there are no tools available for conditional tagging of metabotropic ACh receptors. In the future, additional FlpTag lines can potentially answer this question.

5) The authors also assess the distribution of receptors in axons. The Discussion is extensive, highlighting, that "the subcellular distribution… highlights its differential functional roles in different parts of these cell types". While, it is clear that the receptors are indicative of different connections, the experiments shown in this study do not provide any functional insights, in particular how inhibitory input at the dendritic level and again at the axonal level could be integrated during information processing of T4/T5 neurons.

In general, the objective of the paper was to develop methods for studying receptor distributions which lay the foundation for functional investigations. As for the example of GluCla in T4/T5: According to RNAseq studies GluCla is expressed in T4 and T5 neurons. We found that this receptor is localized to dendrites and terminals in T4 and only terminals in T5. Without this knowledge functional experiments, e.g. RNAi knockdown of GluCla in T4 vs. T5 would disturb completely different synaptic mechanisms in these cells (dendritic vs. axonal inhibition) and the interpretation of the results would be challenging. Hence, we propose that in general it would always be beneficial to first study which receptors are expressed and how they are localized before functional experiments are undertaken. Clearly, knockdown or knock-out experiments of the described receptors will be the next important step for our future studies.

6) Subsection “Methods for investigation of receptor distributions”, the manuscript discusses the possibility of using their construct also for Crispr/Cas9 mediated insertion by a HDR cassette, referring to supplementary files solely with sequences. This is an important aspect and would benefit from a more detailed presentation and evidence for functionality to support the claim and the technical emphasis of the study in general.

We added more information to the Discussion.

Reviewer #2:The tools the authors developed expand the repertoire of similar reagents already available to access neurotransmitter receptor expression in the fly. Additionally, the novel FlpTag tool allows to endogenously tag any gene of interest in a cell type specific manner expanding its usefulness to different molecular questions in various fields. The paper is well written, the results are solid and well supported but fall short. The results are limited to the validation of a previously published UAS-Da7::GFP line, two UAS lines the authors generated (UAS-GluCla::GFP and UAS-Rdl::GFP), the novel FlpTag tool, and to the description of the studied neurotransmitter receptors expression in the T4/T5 circuit. More specifically, the authors show that the GluCla, Rdl and Da7 distribution in T4 and T5 dendrites matches the known EM-reconstructed synapse distribution and neurotransmitter identity of upstream input neurons and it is compatible with one of several algorithmic models that describe how motion direction-selective responses in T4/T5 neurons is established. They further show a previously unknown expression of the glutamate receptor GluCla in T4/T5 axon terminals and discuss its possible role. As such, I find the manuscript in its current form better suited for a Tools and Resources article. I would support such publication.Reviewer #3:The paper uses a combination of novel genetic and classical immunohistochemical tools to pinpoint the receptor identities and their spatial distributions on T4/5 cells in the fruit fly visual system. In general, spatially offset receptor distributions here are known key ingredients to motion computation and understanding them in depth is certainly a useful thing to do. The paper extends previous results based on transcriptomics and EM (and other approaches) to lay out very nicely which receptors sit in which part of the dendrites. Overall, the results are very much in line with previous work and present an important consolidation of the literature. The paper is generally well written and quite clear. Accordingly, I support publication in eLife in principle.However, it may be useful for the authors to consider some of the below1) Stats + quantification: most statistics (all except Figure 2—figure supplement 1D) in the manuscript are summarised by bar plots with an error bar. This could do with an upgrade. E.g. convert to box/whisker-plots (or similar) and add raw data as a scatter in the background. Also, I notice the authors used t-tests in a few places, but I cannot see a test for normal distribution in the data (and since the raw data is not shown I also cannot eyeball this).

Thank you for the helpful suggestion; we updated to box plots with a scatter in the background.

2) Related, the histograms (e.g. 2E, 4D, 5D, 6B,D) all seem to represent a mean across a bunch of experiments. These need some form of quantification of variance. A simple option could be confidence interval shadings. Moreover, they could do with stats. e.g. when comparing 2 distributions, one could fit a Generalized Additive Model (GAM) to pinpoint (i) if and (ii) where they are different (or something similarly telling).

Likewise, histograms in Figures 2E, 4D, 5D have been updated to show confidence interval shading. We left the summary Figure 6B and D without ci shading for easier visualization.

Related to the above, I would have expected the receptor distributions to be quantified over 2D dendritic area, rather than "just" the linear distance along the dendrite's major axis. I appreciate that this is the axis that matters for the computation, but since the data is available in 2D, why not use it? Maybe there is additional nuance to be gained? The authors could compute the mean density map of receptors and plot it next to their example images. This would presumably also aid to more intuitively highlight the spatial offsets? It might make for a nice summary figure as well.

This is a good suggestion and we initially plotted the distribution of the dendritic area in 2D. However, we found that there is little to be gained from adding this second dimension. Thus, we opted for a 1D representation for simplicity and easier visualization.

Reviewer #4:Fendl et al. develop new labeling strategies to assess the asymmetric distribution of neurotransmitter subunits for glutamate, GABA, and Ach in the visual system. In particular, they develop ways to visualize GluClα, Dα7, and Rdl in motion-sensing T4/T5 neurons in *Drosophila*. The authors do a great job of carefully controlling for possible artifacts of overexpression in the channels they overexpress and show compelling evidence of distinct localization patterns of these three receptors in dendrites of the T4/T5 neurons. Overall, the manuscript is well written and presents both technical and biological advances, although both are somewhat incremental in significance. On the technical side, the development of the "FlpTag" tool for conditional labeling of proteins uses well-established technology and similar approaches have already been utilized. In addition, only a single receptor, GluClα, was actually successfully targeted for this FlpTag labeling. Beyond this technical achievement, the authors essentially confirm the localization of the three neurotransmitter receptors expected by the distribution of EM-reconstructed synapses of the relevant input neurons in the visual system. Thus, the contribution here of generating UAS-receptor-tags and the GluClα FlpTag to confirm putative receptor distributions appears to be an incremental contribution to the field. That being said, there are some additions that can improve the impact and rigor of this study.1) Technical development: The authors used their FlpTag approach on GluClα and did not establish a toolkit targeting other neurotransmitter receptor subtypes. It's not clear why this strategy wasn't used for other receptors, at least for the two other receptors characterized in this study (ACh Dα7 and Rdl).

We included more FlpTag lines and discussed the underlying caveats for some cases.

Furthermore, a recent publication in eLife demonstrated that GluClα can be tagged conditionally as well (Molina-Obando et al., 2019), so it's not clear how important this reagent will prove to be to the field by targeting just this receptor.

We are not sure what the reviewer is referring to. There is no conditional tagging tool for GluClα published in Molina-Obando et al., 2019. They merely used the already existing MiMIC GFSTF line to conclude that GluClα is broadly expressed in the visual system in the fly (Figure 4). This MiMIC line is a pan-neuronal, endogenous GFP-tagged allele of GluClα, but it is not conditional and it cannot be used to assign expression of this receptor to a specific cell-type, as we pointed out in our study. Furthermore, they created a GluClα-FlpStop line and a GluClα-allele insensitive to picrotoxin for functional investigations, but again no conditional labeling strategy for this receptor (Figure 5 and Figure 7). Taken together, to the best of our knowledge, there exists no other tool for conditional tagging of GluClα.

An expanded toolkit, akin to the one tagging all neurotransmitter receptors in the fly genome at their endogenous locus using T2A-Gal4 knock-in (Kondo et al., 2020), would be a great addition for the field but it is understood this is beyond the scope of the investigators interest in this study.

We agree that an expanded toolkit with conditional tagging lines for all neurotransmitter receptors would be great. However, this is not a trivial endeavor due to several reasons as we discuss in subsection “Expansion of the FlpTag toolbox”.

2) Biological insights: It is commendable the extent the authors went to carefully control for possible artifacts of mis-localization of the three neurotransmitter receptors due to overexpression of the straightforward UAS-receptor-tags. Beyond overexpression, it is always possible that the tags themselves may alter receptor trafficking, biochemical interactions, or integration into functional receptors. Can the authors comment in more detail on controls used to determine the tags do not impact receptor localization or function?

We controlled for the potential receptor mis-localization of UAS-lines by comparing the distribution to additional endogenous (e.g. MiMIC-, TG4-lines) lines.

At least for the three receptors examined in this study, overexpression doesn't appear to lead to mis-localization or apparent artifacts. However, in terms of the biological significance, it's not clear how much is learned. Essentially, all the receptors mark synapses in areas they were suspected to be based on the neuronal inputs, and functional imaging or electrophysiology/pharmacology would be necessary to validate the ideas of how these receptors and their presumed physiological impacts actually shape signal processing in the T4/T5 neurons.

It is true that “the receptors mark synapses in the areas they were suspected to be based on the neuronal inputs”. However, we believe that this remains speculative until a sound proof is provided. It is not trivial to develop the tools and investigate the subcellular distributions of the receptors as we did in our work. Furthermore, as we pointed out, there are numerous receptor subunits for the different transmitter classes and it was not clear which subunits would localize to the different synaptic sites. Hence, these investigations are important in order to define for instance the sign of the synaptic connection.

3) Although it is important to define which receptor subtypes exist at particular synapses, the experiments presented here by the authors do not really establish the receptors that function at the dendrites in the T4/T5 neurons. First, as the authors point out, it is a mystery what Ach receptor subtype(s) actually exist and function at dendrites in this system, and even whether Dα7 is playing a functional role. Second, while GluClα appears to be present at glutamatergic synapses, can the authors rule out the possibility of excitatory glutamate receptors also being present in T4/T5 neurons? It seems plausible that some of the kainate, AMPA, and/or NMDA type glutamate receptors encoded in the fly genome could also exist in this system, and if so these would have major impacts on the assumptions of signal processing proposed by the authors. Third, and for similar reasons, the findings on GABA receptors may be limited. Finally, also as pointed out by the authors, there are metabotropic receptors that could be expressed and play important roles that remain at present unknown.

We agree that the insights gained with the UAS-Da7 line might be limited as we discussed in our manuscript. However, this validations and controls are still valuable and have to be made for every new line used. For instance, we showed that under UAS-driven expression Da7::GFP is localizing only to cholinergic synapses, still constituting an important marker for postsynaptic cholinergic sites.

Furthermore, it is true that it could always be the case that other receptor subunits might be present at the specific synaptic sites as well. As for glutamatergic synapses in T4 dendrites, this is very unlikely since the expression values for the other glutamate receptors are very low (Davis et al., 2020; Pankova et al., 2016). In general, our study is just the starting point of these types of investigation and additional FlpTag lines will help to discover the full repertoire of ion channels in T4/T5 neurons in the future.